# Integrative and comparative genomic analyses identify clinically relevant pulmonary carcinoid groups and unveil the supra-carcinoids

N. Alcala [iD] et al.[#]

The worldwide incidence of pulmonary carcinoids is increasing, but little is known about their molecular characteristics. Through machine learning and multi-omics factor analysis, we compare and contrast the genomic profiles of 116 pulmonary carcinoids (including 35 atypical), 75 large-cell neuroendocrine carcinomas (LCNEC), and 66 small-cell lung cancers. Here we report that the integrative analyses on 257 lung neuroendocrine neoplasms stratify atypical carcinoids into two prognostic groups with a 10-year overall survival of 88% and 27%, respectively. We identify therapeutically relevant molecular groups of pulmonary carcinoids, suggesting DLL3 and the immune system as candidate therapeutic targets; we confirm the value of *OTP* expression levels for the prognosis and diagnosis of these diseases, and we unveil the group of supra-carcinoids. This group comprises samples with carcinoid-like morphology yet the molecular and clinical features of the deadly LCNEC, further supporting the previously proposed molecular link between the low- and high-grade lung neuroendocrine neoplasms.

According to the WHO classification from 2015[1] and a recent IARC-WHO expert consensus proposal[2], pulmonary carcinoids are low-grade typical and intermediate-grade atypical well-differentiated lung neuroendocrine tumours (LNETs) that belong to the group of lung neuroendocrine neoplasms (LNENs), which also includes the high-grade and poorly differentiated small-cell lung cancer (SCLC) and large-cell neuroendocrine carcinomas (LCNEC). Pulmonary carcinoids are rare malignant lesions, annual incidence of which has been increasing worldwide, especially at the advanced stages[3]. Pulmonary carcinoids account for 1–2% of all invasive lung malignancies: typical carcinoids exhibit good prognosis, although 10-23% metastasise to regional lymph nodes, resulting in a 5-year overall survival rate of 82–100%. The prognosis is worse for atypical carcinoids, with 40–50% presenting metastasis, reducing the 5-year overall survival rate to 50%.

Contrary to pulmonary carcinoids, most of which are eligible for upfront surgery at the time of diagnosis[3], LCNEC and SCLC require upfront aggressive, multimodal treatment for most of the patients. Owing to these differences in clinical management and prognosis, the accurate diagnosis of these diseases is critical. However, there is still no consensus on the optimal approach for their differential diagnosis;[2] the current criteria, based on morphological features and immunohistochemistry, are imperfect and inter-observer variations are common, especially when separating typical from atypical carcinoids[4], as well as atypical carcinoids from LCNEC in small biopsies[5]. Ki67 protein immune-reactivity has been suggested as a good marker of prognosis in LNENs as a whole, and for the differential diagnosis between carcinoids and SCLC[6,7], whereas this marker does not faithfully follow the defining histological criteria of typical and atypical carcinoids[4]. The difficulties in finding good markers to separate these diseases might be due to the limited amount of comprehensive genomic studies available for SCLC, LCNEC, and typical carcinoids, and the complete lack of such studies for atypical carcinoids[8]. In addition, such studies would also be needed to validate the recent proposed molecular link between pulmonary carcinoids and LCNEC[9,10].

In this study, we provide a comprehensive overview of the molecular traits of LNENs—with a particular focus on the understudied atypical carcinoids—in order to identify the mechanisms underlying the clinical differences between typical and atypical carcinoids, to understand the suggested molecular link between pulmonary carcinoids and LCNEC, and to find new candidates for the diagnosis and treatment of these diseases.

## Results

**Data.** We have generated new data (genome, exome, transcriptome, and methylome) for 63 pulmonary carcinoids (including 27 atypical) and 20 LCNEC. In order to perform comparative analyses, we have reanalysed published data for 74 pulmonary carcinoids[11], 75 LCNEC[12], and 66 SCLC[13,14]. Taken together, we have performed multi-omics integrative analyses on 116 pulmonary carcinoids (including 35 atypical), 75 LCNEC, and 66 SCLC (Supplementary Fig. 1 and Supplementary Data 1).

**Molecular groups of pulmonary carcinoids and LCNEC.** We performed an unsupervised analysis of the expression and methylation data of the LNENs (i.e., 110 pulmonary carcinoids and 72 LCNEC) using the Multi-Omics Factor Analysis implementation of the group factor analysis statistical framework (Software MOFA)[15] (MOFA LNEN; Fig. 1a and Supplementary Figs. 2 and 3). We identified five latent factors explaining more than 2% of the variance in at least one data set, and among them, three latent factors provided consistent groups of samples with similar expression and methylation profiles (i.e., clusters). MOFA latent factors one (LF1) and two (LF2) explained a total of 45% and 34% of the variance in methylation and expression, respectively, and were both associated with survival (Supplementary Fig. 4). Using consensus clustering on these two latent factors (which explained most of the variation and thus carried most of the biological signal; Supplementary Figs. 5–7 and Supplementary Data 2–3), we identified three clusters, each of them enriched for samples of one of the three histopathological types (Fig. 1a). Cluster Carcinoid A was enriched for typical carcinoids (75%; Fisher's exact test p-value $< 2.2 \times 10^{-16}$); cluster Carcinoid B was enriched for atypical carcinoids (54%; Fisher's exact test p-value $< 2.2 \times 10^{-16}$) and male patients (79%; Fisher's exact test p-value $= 1.6 \times 10^{-9}$); and cluster LCNEC included 92% of the histopathological LCNEC (Fisher's exact test p-value $< 2.2 \times 10^{-16}$). Note that clustering based on LF1 to LF5, weighted by their proportion of variance explained, leads to the exact same clusters (Supplementary Fig. 8).

To assess whether the current histopathological classification could be improved by the combination of molecular and morphological characteristics, we undertook a machine-learning (ML) analysis. To do so, we combined the predictions from two independent random forest classifications, based on only-expression or only-methylation data. Using two independent models allowed the inclusion of samples for which only one of these data sets was available, thus maximising the power of subsequent analyses (Fig. 1b and Supplementary Fig. 9 for an alternative analysis based on both 'omic data sets simultaneously, but restricted to fewer samples). In order to avoid overfitting the data, we performed a leave-one-out cross-validation, with feature filtering and normalisation learned from the training set and applied to the test sample. To identify intermediate profiles, we defined a prediction category (unclassified) for samples that had a probability ratio between the two most probable classes close to one. We present in Fig. 1b the results for a cutoff ratio of 1.5, and show in Supplementary Fig. 10 the robustness of our results with regard to this ratio. Ninety-six per cent of the carcinoids predicted as typical by the ML were in cluster Carcinoid A (Fig. 1a). Similarly, the majority of ML-predicted atypical carcinoids (87%) belonged to cluster Carcinoid B.

We selected the ML-prediction groups with >10 samples (gathering the unclassified samples in one single group) and compared their overall survival using Cox's proportional hazard model (coloured groups in Fig. 1b). The machine learning trained on the histopathology stratified atypical carcinoids into two prognostic groups: the good-prognosis group (atypical reclassified as typical, in pink in Fig. 1b, c) with a 10-year overall survival similar to that of samples confirmed by ML as typical carcinoids (in black in Fig. 1b, c; 88% and 89%, respectively; Wald test p-value = 0.650); and the bad-prognosis group (atypical predicted as atypical, in red in Fig. 1b, c) with a 10-year overall survival similar to that of samples confirmed by ML as LCNEC (in blue in Fig. 1b, c; 27% and 19% respectively; Wald test p-value = 0.574; see also Supplementary Fig. 11). Machine-learning analyses based on other features -combined expression and methylation data (Supplementary Fig. 9), MOFA latent factors (Supplementary Fig. 12A), and Principal component analyses (PCA) principal components explaining more than 2% of the variance (Supplementary Fig. 12B)- led to qualitatively similar results.

**Atypical carcinoids with LCNEC molecular characteristics.** Six atypical carcinoids clustered with LCNEC in the MOFA LNEN (supra-carcinoids; Fig. 1a). Consistent with this clustering, this group displayed a survival similar to the other samples in the LCNEC cluster (10-year overall survival of 33% and 19%,

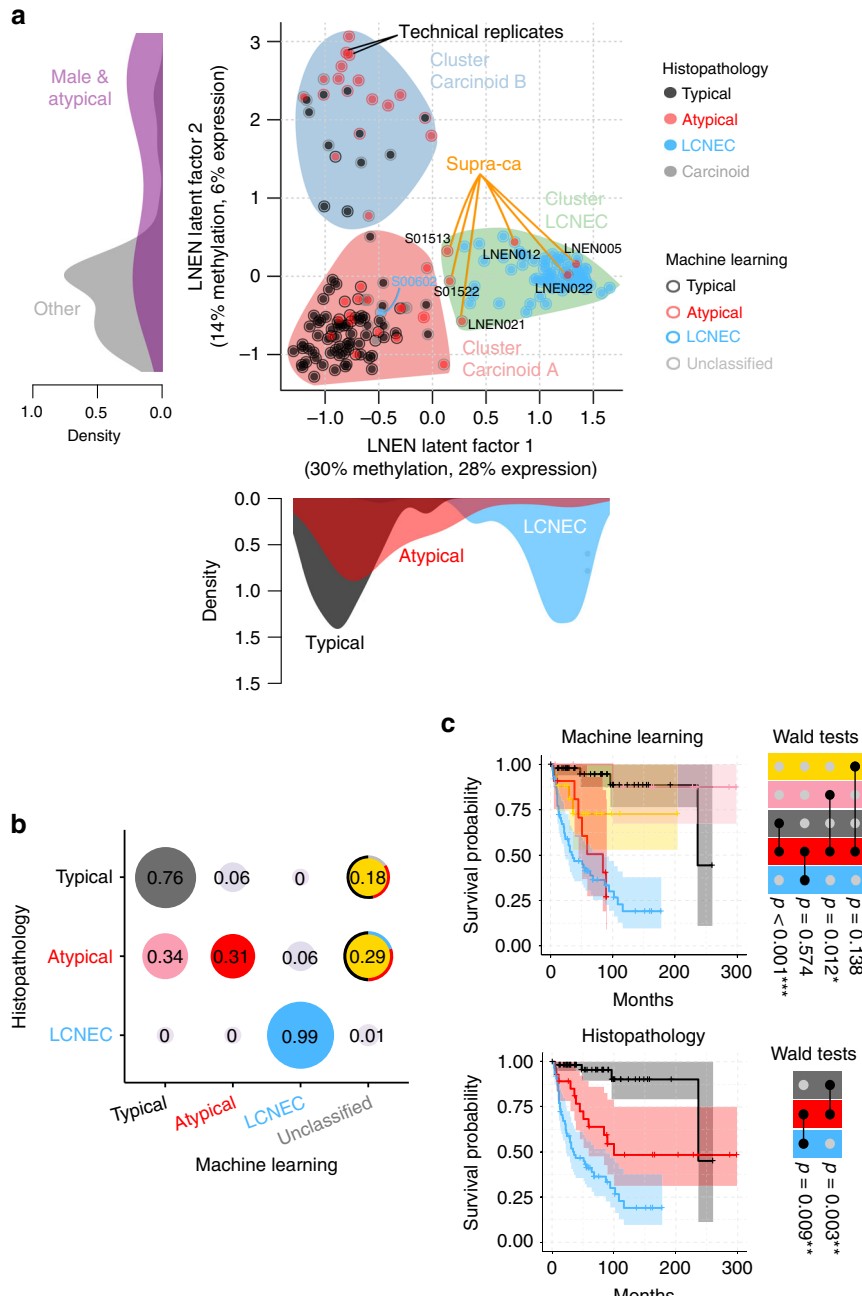

**Fig. 1** Multi-omics (un)supervised analyses of lung neuroendocrine neoplasms. **a** Multi-omics factor analysis (MOFA) of transcriptomes and methylomes of LNEN samples (typical carcinoids, atypical carcinoids, and LCNEC). Point colours correspond to the histopathological types; coloured circles correspond to predictions of histopathological types by a machine learning (ML) algorithm (random forest classifier) outlined in **b**; filled coloured shapes represent the three molecular clusters identified by consensus clustering. The density of clinical variables that are significantly associated with a latent factor (ANOVA q-value < 0.05) are represented by kernel density plots next to each axis: histopathological type for latent factor 1, sex and histopathological type for latent factor 2. **b** Confusion matrix associated with the ML predictions represented on **a**. The different colours highlight the prediction groups considered in the survival analysis and the colours for machine learning are consistent between panel **b** and upper panel **c**. Black represents typical carcinoids predicted as typical, pink represents atypical carcinoids predicted as typical, red represents atypical carcinoids predicted as atypical, and blue represents LCNEC samples predicted as LCNEC. For the unclassified category, the most likely classes inferred from the ML algorithm are represented by coloured arcs (black for typical, red for atypical, blue for LCNEC, and light grey for discordant methylation-based and expression-based predictions). **c** Kaplan–Meier curves of overall survival of the different ML predictions groups (upper panel) and histopathological types (lower panel). Upper panel: colours of predicted groups match panel **b**. Lower panel: black-typical, red-atypical, blue-LCNEC. Next to each Kaplan–Meier plot, matrix layouts represent pairwise Wald tests between the reference group and the other groups, and the associated p-values; $0.01 \leq p < 0.05$, $0.001 \leq p < 0.01$, and $p < 0.001$ are annotated by one, two, and three stars, respectively. Data necessary to reproduce the figure are provided in Supplementary Data 1

respectively; Wald test p-value = 0.574; Fig. 2a). The observed molecular link appears to be between supra-carcinoids and LCNEC rather than with SCLC, as shown by PCA and MOFA including expression data for 51 SCLC (Supplementary Figs. 6 and 13, respectively).

These samples originated from three different centres (two from each), and included two previously published samples (S01513 and S01522)[11], implying that this observation is unlikely to be the result of a batch effect. The limited number of supra-carcinoids did not allow to explore aetiological links; however, it is of note that one of them (LNEN005) belonged to a patient with professional exposure to asbestos (which is known to cause mesothelioma)[16] (Table 1), and the tumour harboured a splicing BAP1 somatic mutation (a gene frequently altered in mesothelioma)[17]. This sample showed the highest mutational load (37 damaging somatic mutations; Supplementary Data 4). Gene set enrichment analyses (GSEA) of mutations in the hallmarks of cancer gene sets[18,19], showed a significant enrichment for the hallmark evading growth suppressor (q-value = 0.0213; Fig. 2b and Supplementary Data 5), while the hallmark genome instability and mutation was significant only at the 10% false discovery rate (FDR) threshold (q-value = 0.0970; Fig. 2b and Supplementary Data 5). We had access to the Haematoxylin and Eosin (H&E) stain for three of these supra-carcinoids, on which the pathologists discarded misclassifications with LCNEC, SCLC, or mesothelioma in the case of the asbestos-exposed BAP1-mutated sample (Fig. 2c and Table 1).

While generally similar to LCNEC, and albeit based on small numbers, the supra-carcinoids appeared to have nonetheless some distinct genomic features based on genome-wide expression and methylation profiles (Fig. 2d). Supra-carcinoids displayed higher levels of immune checkpoint genes (both receptors and ligands; Fig. 2e), and also harboured generally higher expression levels of MHC class I and II genes (Fig. 2e and Supplementary Fig. 14). Interestingly, the interferon-gamma gene—a prominent immune-stimulator, in particular of the MHC class I and II genes—also showed high-expression levels in these samples (Supplementary Fig. 14). The differences in immune checkpoint gene expression levels between groups were not explained by the amount of infiltrating cells, as estimated by deconvolution of gene expression data with software quanTIseq (Fig. 2f, left panel). However, supra-carcinoids contained the highest levels of neutrophils (greater than the 3rd quartile of the distributions of neutrophils in the other groups; Fig. 2f, right panel). Permutation tests showed that these levels were significantly higher than in other carcinoid groups and in SCLC, but not than in LCNEC (Supplementary Fig. 15). Concordantly, GSEA showed that MOFA LNEN LF1 (separating LCNEC and supra-carcinoids from the other carcinoids) was significantly associated with neutrophil chemotaxis and degranulation pathways (Supplementary Data 6). By contrast, no such association was observed in the MOFA performed only on carcinoids and SCLC samples (Supplementary Figs. 6C and 13C and Supplementary Data 6).

**Mutational patterns of pulmonary carcinoids**. In a previous study, mainly including typical carcinoids, we detected MEN1, ARID1A, and EIF1AX as significantly mutated genes[11]. We also found that covalent histone modifiers and subunits of the SWI/SNF complex were mutated in 40% and 22.2% of the cases, respectively. Genomic alterations in these genes and pathways were also seen in the new samples included in this study (Fig. 3a, Supplementary Fig. 16, and Supplementary Data 4). Apart from the above-mentioned genes, ATM, PSIP1, and ROBO1 also showed some evidence, among others, for recurrent mutations in pulmonary carcinoids (Fig. 3a). In addition to point mutations

and small indels, the ARID2, DOT1L, and ROBO1 genes were also altered by chimeric transcripts (Fig. 3b). MEN1 was also inactivated by genomic rearrangement in a carcinoid sample with a chromothripsis pattern affecting chromosomes 11 and 20 (Fig. 3c). The full lists of somatically altered genes, chimeric transcripts, and genomic rearrangements are presented in Supplementary Data 4, 7, and 8, respectively. Of note, MEN1 mutations were significantly associated with the atypical carcinoid histopathological subtype (Fisher's exact test p-value = 0.0096), as well as MOFA LNEN LF2.

**Altered pathways in pulmonary carcinoids**. The third latent factor from the MOFA LNEN accounted for 8% and 6% of the variance in expression and methylation, respectively, but unlike LF1 and LF2, LF3 was not associated with patient survival (Supplementary Fig. 4). The molecular variation explained by LF3 appeared to capture different molecular profiles within cluster Carcinoid A (Supplementary Fig. 13B). We therefore undertook an additional MOFA restricted to pulmonary carcinoid samples only (MOFA LNET; Fig. 4a and Supplementary Fig. 17). This MOFA identified five latent factors that explained at least 2% of the variance in one data set. As expected, the first two latent factors of the MOFA LNET were highly correlated with LF2 and LF3 from the MOFA LNEN, respectively, (Pearson correlation >0.96; Supplementary Fig. 13B), and explained 41% and 35% of the variance in methylation and expression, respectively. Integrative consensus clustering using LF1 and LF2 of the MOFA LNET identified three clusters (Supplementary Fig. 18): cluster Carcinoid A1 and cluster Carcinoid A2, that together correspond to the samples in cluster Carcinoid A of the MOFA LNEN, plus the supra-carcinoids; and cluster Carcinoid B (as for the clustering of LNEN samples, a clustering based on LF1-LF5 weighted by their proportion of variance explained, led to the exact same clusters; Supplementary Fig. 8). LF2 was associated with age, with cluster Carcinoid A1 enriched for older patients ((60, 90] years old) and cluster Carcinoid A2 enriched for younger patients ((15, 60] years old).

We applied GSEA to identify the pathways associated with the different latent factors. We found significant associations with the immune system and the retinoid and xenobiotic metabolism pathways (Supplementary Data 6). Numerous Gene Ontology (GO) terms and KEGG pathways were related to the immune system, immune cell migration, and infectious diseases. The GO terms and KEGG pathways related to immune cell migration included leucocyte migration, chemotaxis, cytokines, and interleukin 17 signalling. In particular, the expression of all β-chemokines (including CCL2, CCL7, CCL19, CCL21, CCL22, known to attract monocytes and dendritic cells)[20] (Supplementary Data 6), and all CXC chemokines (such as IL8, CXCL1, CXCL3, and CXCL5, known to attract neutrophils)[21], were positively correlated with MOFA LNEN LF1 (separating pulmonary carcinoids from LCNEC) and negatively correlated with MOFA LNET LF2 (separating clusters Carcinoid A1 and A2).

The different LNET clusters did not differ in their total amounts of estimated proportions of immune cells, but they did differ in their composition (Supplementary Fig. 19): cluster Carcinoid A (particularly A1) was significantly enriched in dendritic cells, and cluster Carcinoid B, in monocytes (Fig. 4b, upper panel). As monocytes can differentiate into dendritic cells in a favourable environment[22], we assessed the levels of LAMP3 and CD1A dendritic-cells markers[23], and found that samples in cluster Carcinoid A1 presented high-expression levels of these genes (Fig. 4b, lower panel), implying that this cluster was indeed enriched for dendritic cells. We pursued this further by assessing

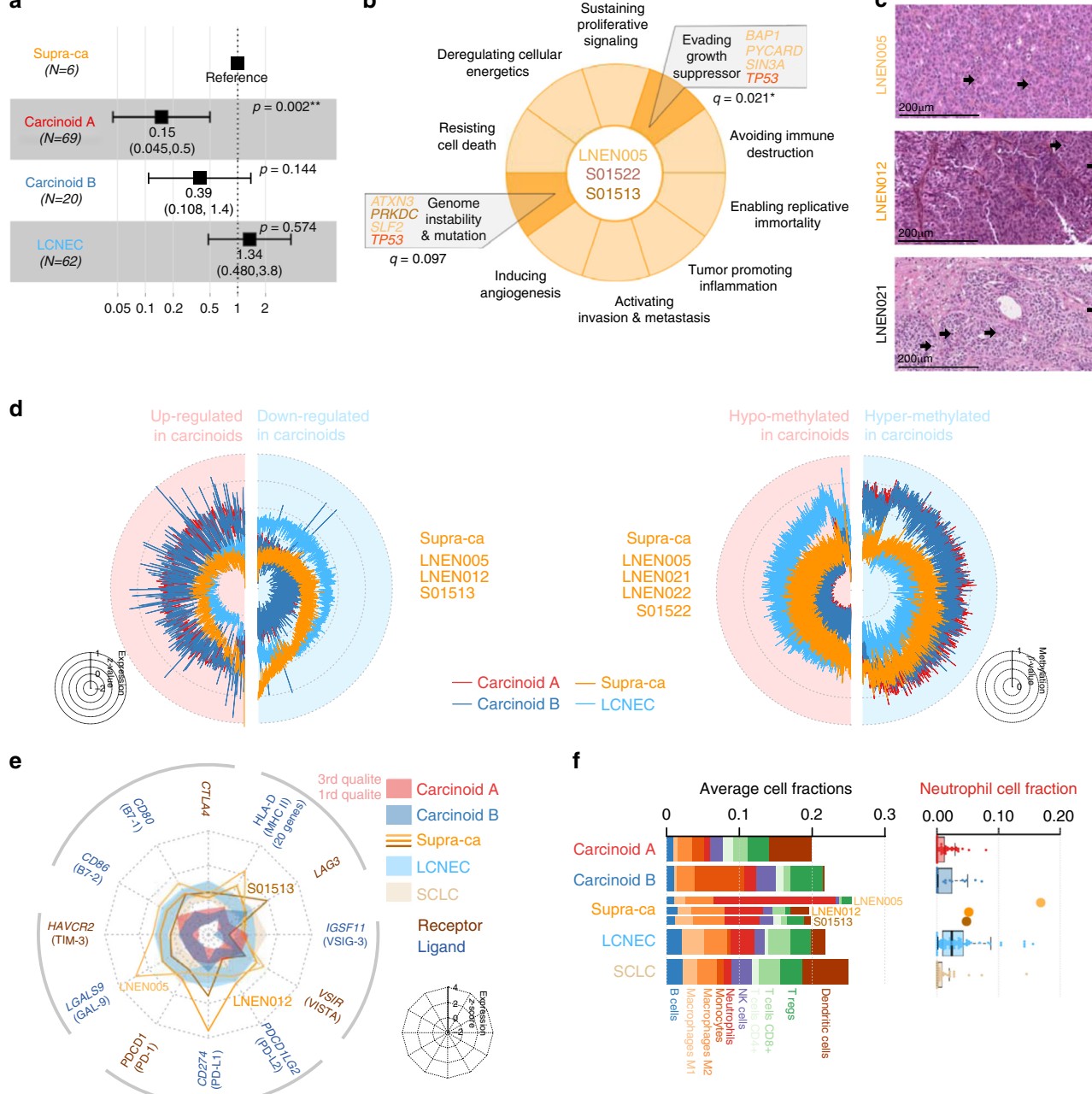

**Fig. 2** Molecular characterisation of supra-carcinoids. **a** Forest plot of hazard ratios for overall survival of the supra-carcinoids, compared to Carcinoid A and B, and LCNEC. The number of samples (*N*) in each group is given in brackets. The black box represent estimated hazard ratios and whiskers represent the associated 95% confidence intervals. Wald test *p*-values are shown on the right. **b** Enrichment of hallmarks of cancer for somatic mutations in supra-carcinoids. Dark colours highlight significantly enriched hallmarks at the 10% false discovery rate threshold; corresponding mutated genes are listed in the boxes, and enrichment *q*-values are reported below. **c** Hematoxylin and Eosin (H&E) stains of three supra-carcinoids. In all cases, an organoid architecture with tumour cells arranged in lobules or nests, forming perivascular palisades and rosettes is observed; original magnification x200. Arrows indicate mitoses. **d** Radar charts of expression and methylation levels. Each radius corresponds to a feature (gene or CpG site), with low values close to the centre and high values close to the edge. Coloured lines represent the mean of each group. Left panel: expression *z*-scores of genes differentially expressed between clusters Carcinoid A and LCNEC or between Carcinoid B and LCNEC. Right panel: methylation *β*-values of differentially methylated positions between Carcinoid A and LCNEC clusters or between Carcinoid B and LCNEC clusters. **e** Radar chart of the expression *z*-scores of immune checkpoint genes (ligands and receptors) of each group. **f** Left panel: average proportion of immune cells in the tumour sample for each group, as estimated from transcriptomic data using software quanTIseq. Right panel: boxplot and beeswarm plot (coloured points) of the estimated proportion of neutrophils, where centre line represents the median and box bounds represent the inter-quartile range (IQR). The whiskers span a 1.5-fold IQR or the highest and lowest observation values if they extend no further than the 1.5-fold IQR. Data necessary to reproduce the figure are provided in Supplementary Data 1, 4, 5, 12, 17, and in the European Genome-phenome Archive

**Table 1 Characteristics of supra-carcinoids**

| | LNEN005 | LNEN012 | LNEN021 | LNEN022 | S01513 | S01522 |
|---|---|---|---|---|---|---|
| **Classification** | | | | | | |
| Histopathology | Atypical | Atypical | Atypical | Atypical | Atypical | Atypical |
| Morphological characteristics | Carcinoid morph. 2 mitoses/2 mm$^2$ No necrosis | Carcinoid morph. 2 mitoses/2 mm$^2$ No necrosis | LCNEC morph. 4 mitoses/2 mm$^2$ No necrosis | NA | NA | NA |
| Machine learning | LCNEC | LCNEC | Unclassified | Unclassified | Atypical | Unclassified |
| **Clinical data** | | | | | | |
| Sex | Male | Female | Female | Female | Male | Male |
| Age at diagnosis | 80 | 70 | 83 | 58 | 58 | 63 |
| TNM Stage | IB | IIIC | IA1 | IIB | IIIA | IV |
| Overall survival (months) | 144.6 | 111.7 | 29.8 | 36.1 | 59 | 7 |
| **Epidemiology** | | | | | | |
| Smoking status | Former | NA | NA | NA | Never | Current |
| Other known exposure | Asbestos | NA | NA | NA | NA | NA |
| **Multi-omics data** | | | | | | |
| Data available | WES, RNAseq, Epic 850K | RNAseq | Epic 850K | Epic 850K | WGS, RNAseq | WES, Epic 850K |
| Cluster MOFA LNEN | LCNEC | LCNEC | LCNEC | LCNEC | LCNEC | LCNEC |
| Cluster MOFA LNET | Carcinoid A1 | Carcinoid A1 | Carcinoid A1 | Carcinoid A1 | Carcinoid A1 | Carcinoid A1 |
| Selected mutated genes | *JMJD1C, KDM5C, BAP1* | NA | NA | NA | *DNAH17* | *TP53* |
| Mean FPKM of IC genes[a] | 8.12 | 10.32 | NA | NA | 3.15 | NA |
| *MKI67* FPKM | 2.6 | 7.3 | NA | NA | 1.9 | NA |

FPKM refers to Fragments Per Kilobase per Million reads. The median FPKM of immune checkpoint (IC) genes was calculated based on the genes included in Fig. 2e, excluding HLA genes because of their very large expression levels
[a]IC genes median FPKM values for pulmonary carcinoids, LCNEC and SCLC are 1.0, 3.5, and 3.2, respectively

the CD1A protein levels by immunohistochemistry (IHC) in an independent series of pulmonary carcinoids, and found that 60% of them (12 out of 20) were enriched in CDA1-positive dendritic cells, confirming the presence of dendritic cells in a subgroup of pulmonary carcinoids (Fig. 4c and Supplementary Data 9).

Regarding the retinoid and xenobiotic metabolism pathways (e.g., elimination of drugs and environmental pollutants), the main genes driving the correlation with MOFA latent factors were the phase II enzymes involved in glucuronosyl-transferase activity (Supplementary Data 6), but also the phase I cytochrome P450 (CYP) proteins. These pathways were positively correlated with MOFA LNEN LF2 (separating LNEN clusters A and B) and negatively correlated with MOFA LNET LF1 (separating LNET clusters A1 and A2 from cluster B). Indeed, we found that samples in cluster Carcinoid B were characterised by high levels of the CYP family of genes, and a very strong expression of several UDP glucuronosyl-transferases *UGT* genes (median FPKM = 4.6 in *UGT2A3* and 28.1 in *UGT2B* genes; Fig. 4d), which contrasts with the low levels in other carcinoids (median FPKM = 0 for both *UGT2A3* and *UGT2B*; Fig. 4d), LCNEC (median FPKM = 0 and 1.2 for *UGT2A3* and *UGT2B*; Supplementary Fig. 20) and SCLC (median FPKM = 0 and 0.3 for *UGT2A3* and *UGT2B*; Supplementary Fig. 20).

**Molecular groups of pulmonary carcinoids**. We explored the molecular characteristics of each cluster from the MOFA LNET based on their core differentially expressed coding genes (core-DEGs, the expression levels of which defined a given group of samples), corresponding promoter methylation profiles (Fig. 5a and Supplementary Data 10), and their somatic mutational patterns (Figs. 3a and 4a). To achieve this goal, we computed the DEGs in all pairwise comparisons between a focal group and the other groups, and then defined core-DEGs as the intersection of the resulting gene sets. We show in Supplementary Fig. 21 that core-DEGs are almost exclusively a subset of the DEGs between the focal group and samples from all other groups taken together. We correlated the gene expression and promoter methylation data of the core-DEGs to identify genes, which expression could

be mainly explained by their methylation patterns (Fig. 5a). One of the top correlations was found for *HNF1A* and *HNF4A* homeobox genes (Supplementary Fig. 22), which were strongly downregulated in cluster Carcinoid A1 samples (Supplementary Fig. 23). In addition, the promoter regions of these genes also harboured core-DMPs (differentially methylated positions) of cluster Carcinoid A1, indicating that their methylation profile is specific of this cluster (Supplementary Data 11). These two genes have been reported as having a role in the transcriptional regulation of *ANGPTL3*, CYP, and UGT genes[24], and could thus explain the differential expression of these genes between the clusters. Samples in cluster Carcinoid A1 were also characterised by high-expression levels of the delta like canonical Notch ligand 3 (*DLL3*, 75% with FPKM > 1) and its activator the achaete-scute family bHLH transcription factor 1 (*ASCL1*) (Fig. 5a and Supplementary Data 10), similar to SCLC and LCNEC (Fig. 5b); however, the expression levels of NOTCH genes did not differ between the different groups (Supplementary Fig. 24). The supra-carcinoids were negative for *DLL3* expression (Fig. 5b), and had generally high-expression levels of *NOTCH1-3* (Supplementary Fig. 24). We additionally tested the DLL3 protein levels in the aforementioned independent series of 20 pulmonary carcinoids and found 40% (eight out of 20) with relatively high expression of DLL3 (Fig. 4d and Supplementary Data 9), while in the other 12 samples DLL3 was strikingly absent (Fig. 4d and Supplementary Data 9). Furthermore, we found a correlation between the protein levels of DLL3 and CD1A (Pearson test *p*-value = 0.00034; Supplementary Fig. 25), providing additional evidence for the existence of a DLL3+ CD1A+ subgroup of carcinoids. Core-DEGs in cluster Carcinoid A2 included the low levels of *SLIT1* (slit guidance ligand 1; 97% with FPKM < 0.01), and *ROBO1* (roundabout guidance receptor 1; 56% with FPKM < 1) (Fig. 5a, b and Supplementary Data 10). This cluster also contained the four samples with somatic mutations in the eukaryotic translation initiation factor 1A X-linked (*EIF1AX*) gene (Fig. 4a). Concordantly, samples with *EIF1AX* mutations had significantly higher coordinates on the MOFA LNET LF2 (*t*-test *p*-value = 0.0342).

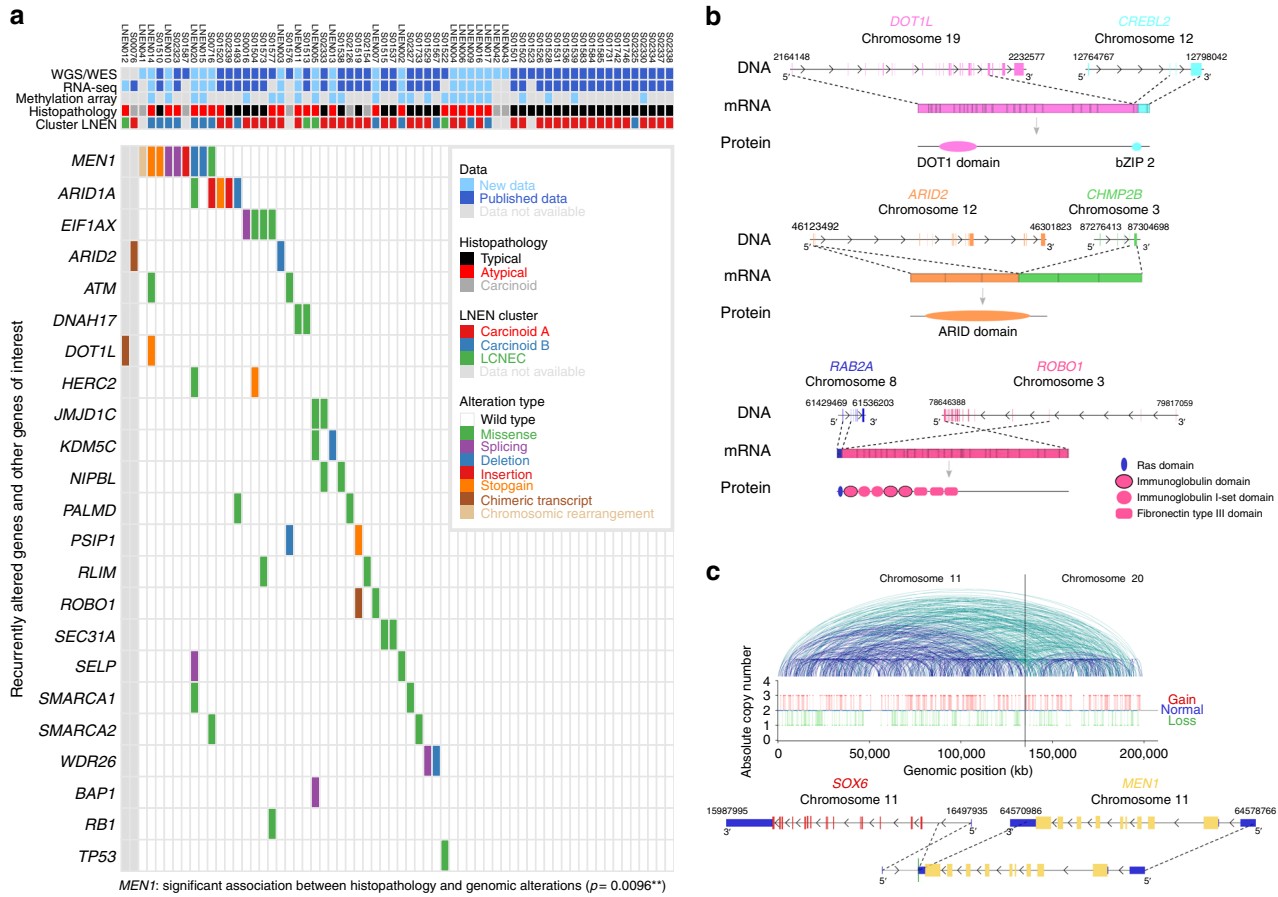

**Fig. 3** Mutational patterns of pulmonary carcinoids. **a** Recurrent and cancer-relevant altered genes found in pulmonary carcinoids by WGS and WES. Fisher's exact test $p$-value for the association between *MEN1* and the atypical carcinoid histopathological subtype is given in brackets; $0.01 \leq p < 0.05$, $0.001 \leq p < 0.01$, and $p < 0.001$ are annotated by one, two, and three stars, respectively. **b** Chimeric transcripts affecting the protein product of *DOT1L* (upper panel), *ARID2* (middle panel), and *ROBO1* (lower panel). For each chimeric transcript the DNA row represents genes with their genomic coordinates, the mRNA row represents the chimeric transcript, and the protein row represents the predicted fusion protein. **c** Chromotripsis case LNEN041, including an inter-chromosomic rearrangement between genes *MEN1* and *SOX6*. Upper panel: copy number as a function of the genomic coordinates on chromosomes 11 and 20; a solid line separates chromosomes 11 and 20. Blue and green lines depict intra- and inter-chromosomic rearrangements, respectively. Lower panel: *MEN1* chromosomic rearrangement observed in this chromotripsis case. Data necessary to reproduce the figure are provided in Supplementary Data 4, 7, and 8

As expected based on Fig. 4d, several UGT genes were core-DEGs of cluster Carcinoid B (Fig. 5a). Also, accordingly with the worse survival of patients in this cluster (Fig. 2a), these samples were also characterised by the expression of angiopoietin like 3 (*ANGPTL3*, 90% with FPKM > 1), and the erb-b2 receptor tyrosine kinase 4 (*ERBB4*, 67% with FPKM > 1) (Fig. 5b). This cluster was also characterised by the universal downregulation of orthopedia homeobox (*OTP*; 90% with FPKM < 1), and NK2 homeobox 1 (*NKX2-1*; 90% FPKM < 1) (Fig. 5b). Interestingly, the SCLC-combined LCNEC sample (S00602) that clustered with the pulmonary carcinoids in the MOFA LNEN (Fig. 1a) was the only LCNEC in our series harbouring high-expression levels of *OTP* (290.26 FPKM vs. 9.89 FPKM for the 2nd highest within LCNEC, the median for LCNEC being 0.22 FPKM). *UGT* genes, *ANGPTL3*, and *ERBB4* were also core-DEGs of cluster B samples when compared to LNEN clusters Carcinoid A and LCNEC (Supplementary Data 12), which indicates that their expression levels also significantly differed from that of LCNEC. Cluster Carcinoid B included all observed *MEN1* mutations, which is consistent with the fact that samples with *MEN1* mutations had significantly lower coordinates on the MOFA LNET LF1 ($t$-test $p$-value = $7 \times 10^{-6}$; Fig. 4a). Nevertheless, mutations in this gene

did not explain the poorer prognosis of this group of samples compared to other LNET (logrank $p$-value > 0.05; Supplementary Fig. 26). To gain some insights into what might be driving the bad prognosis of cluster Carcinoid B samples, we performed a GSEA of mutations in hallmarks of cancer gene sets[18,19]; while clusters Carcinoid A1 and A2 were not enriched for any hallmark of cancer, cluster Carcinoid B was significantly enriched for genes involved in evading growth suppressor, sustaining proliferative signalling, and genome instability and mutation at the 5% FDR (Fig. 5c). We also performed a Cox regression with elastic net regularisation based on the core-DEGs of this cluster; the model selected eight coding genes explaining the overall survival, *OTP* being one of them (Fig. 5d and Supplementary Data 13). Further supporting their prognostic value, we found that the expression of four of these genes was significantly different between the good- and the poor-prognosis atypical carcinoids based on the machine-learning predictions (Fig. 1c, upper panel and Supplementary Fig. 27).

Finally, we also checked the *MKI67* expression levels in the different molecular groups and found relatively low levels in the clusters Carcinoids A1, A2, and B (78% with FPKM < 1) and high levels in the supra-carcinoids (FPKM > 1 in the three samples). As

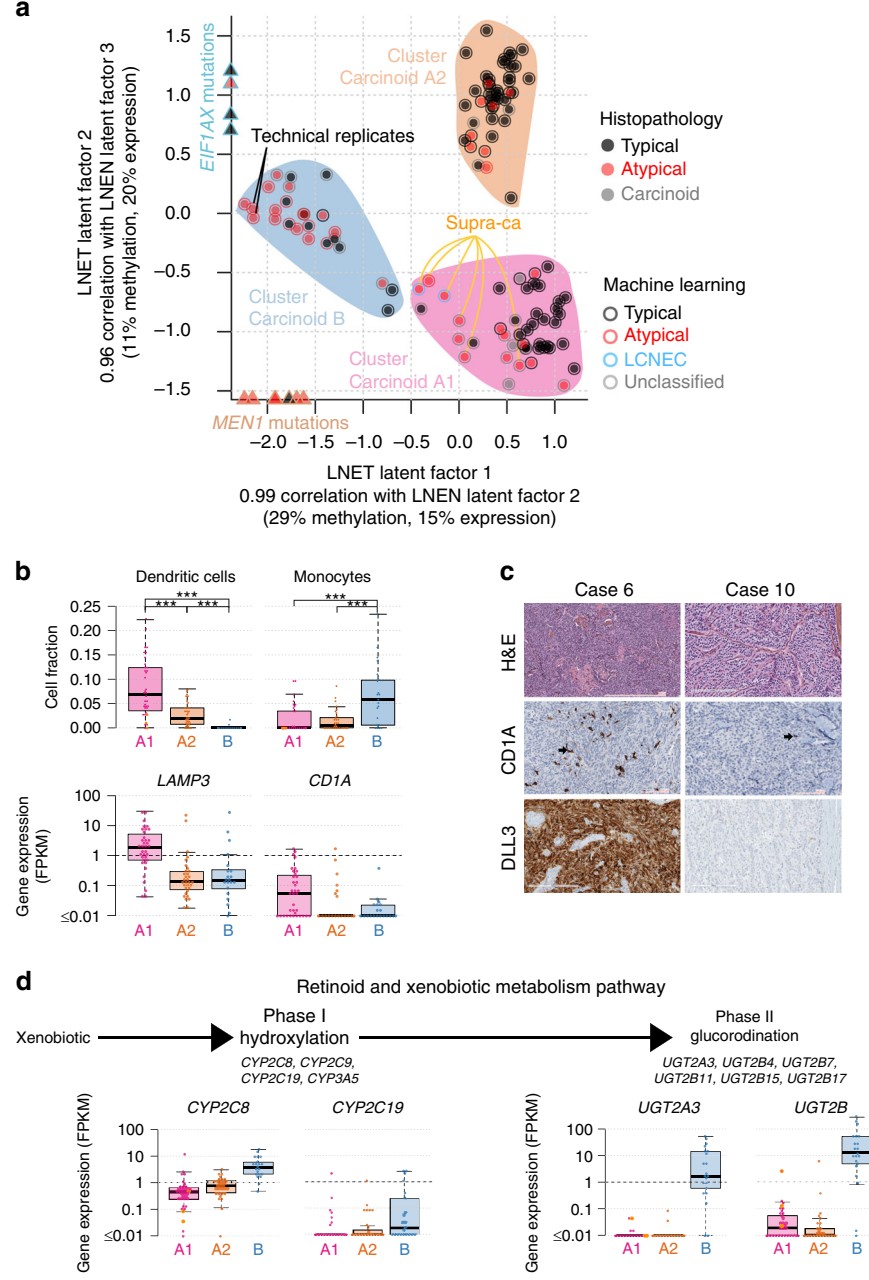

**Fig. 4** Multi-omics unsupervised analysis of lung neuroendocrine tumours. **a** Multi-omics factor analysis (MOFA) of transcriptomes and methylomes restricted to LNET samples (pulmonary carcinoids). Design follows that of Fig. 1a; filled coloured shapes represent the three molecular clusters (Carcinoid A1, A2, and B) identified by consensus clustering. The position of samples harbouring mutations significantly associated with a latent factor (ANOVA q-value < 0.05) are highlighted by coloured triangles on the axes. **b** Upper panel: boxplots of the proportion of dendritic cells in the different molecular clusters (Carcinoid A1, A2, and B) and the supra-carcinoids, estimated from transcriptomic data using quanTIseq (Methods). The permutation test q-value range is given above each comparison: q-value < 0.001 is annotated by three stars. Lower panel: boxplots of the expression levels of *LAMP3* (CDLAMP) and *CD1A*. **c** DLL3 and CD1A immunohistochemistry of two typical carcinoids: case 6 (DLL3+ and CD1A+), and case 10 (DLL3- and CD1A-). Upper panels: Hematoxylin & Eosin Saffron (H&E) stain. Middle panels: staining with CD1 rabbit monoclonal antibody (cl EP3622; VENTANA), where arrows show positive stainings. Lower panels: Staining with DLL3 assay (SP347; VENTANA). **d** Expression levels of genes from the retinoid and xenobiotic metabolism pathway—the most significantly associated with MOFA latent factor 1—in the different molecular clusters. Upper panel: schematic representation of the phases of the pathway. Lower panel: boxplot of expression levels of *CYP2C8* and *CYP2C19* (both from the CYP2C gene cluster on chromosome 10), *UGT2A3,* and the total expression of *UGT2B* genes (from the UGT2 gene cluster on chromosome 4), expressed in fragments per kilobase million (FPKM) units. In all panels, boxplot centre line represents the median and box bounds represent the inter-quartile range (IQR). The whiskers span a 1.5-fold IQR or the highest and lowest observation values if they extend no further than the 1.5-fold IQR. Data necessary to reproduce the figure are provided in Supplementary Data 1, 4, 9, and in the European Genome-phenome Archive

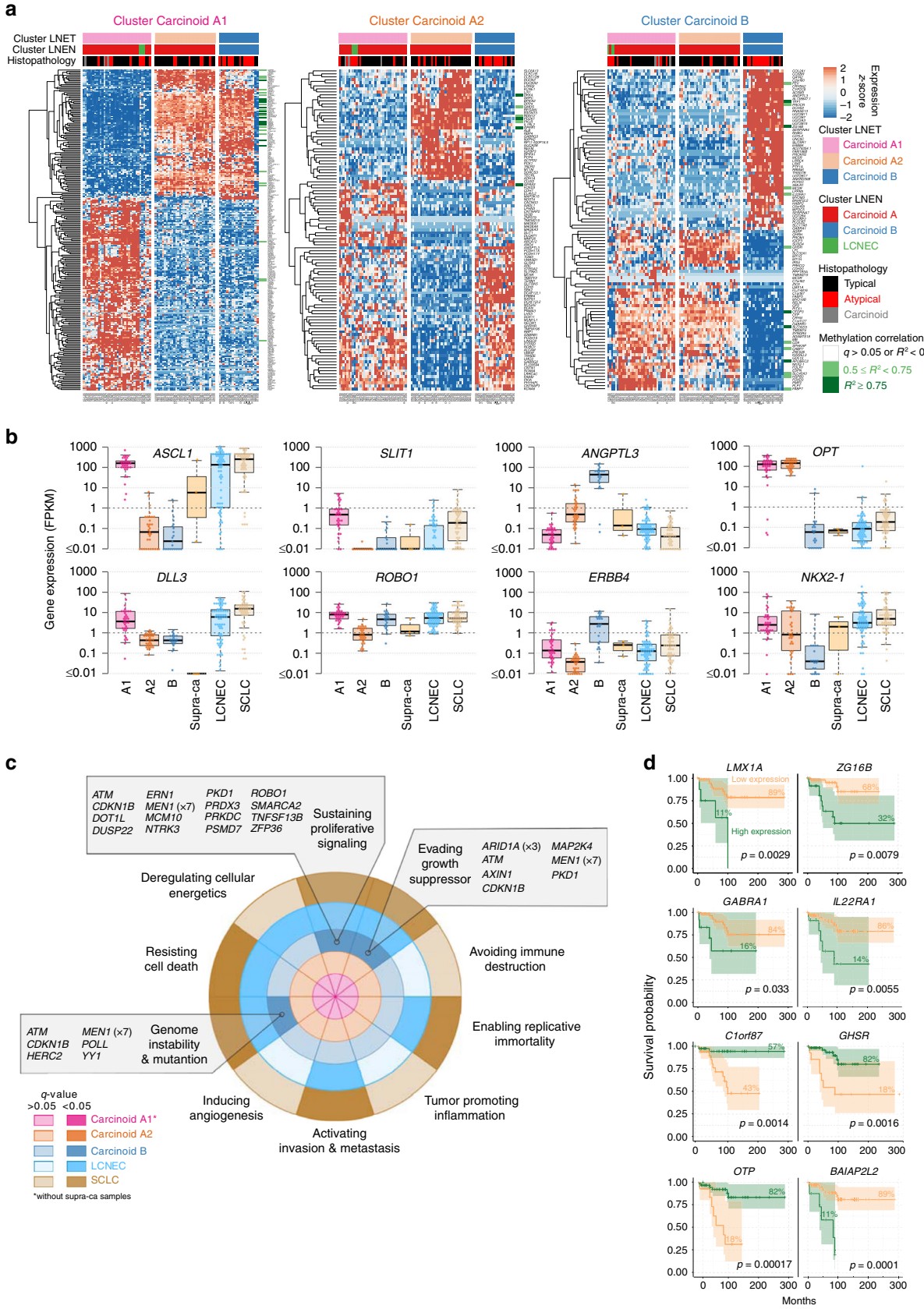

**Fig. 5** Molecular groups of pulmonary carcinoids. **a** Heatmaps of the expression of core differentially expressed genes of each molecular cluster, i.e., genes that are differentially expressed in all pairwise comparisons between a focal cluster and the other clusters. Green bars at the right of each heatmap indicate a significant negative correlation with the methylation level of at least one CpG site from the gene promoter region. The colour scale depends on the range of q-value (q) and squared correlation estimate ($R^2$) of the correlation test. **b** Boxplots of the expression levels of selected cancer-relevant core genes, in fragment per kilobase million (FPKM) units, where centre line represents the median and box bounds represent the inter-quartile range (IQR). The whiskers span a 1.5-fold IQR or the highest and lowest observation values if they extend no further than the 1.5-fold IQR. **c** Characteristic hallmarks of cancer in each molecular cluster (Carcinoid A1 without the supra-carcinoids, A2, and B), LCNEC, and SCLC. Coloured concentric circles correspond to the molecular clusters. For each cluster, dark colours highlight significantly enriched hallmarks (Fisher's exact test q-value < 0.05). The mutated genes contributing to a given hallmark are listed in the boxes. Recurrently mutated genes are indicated in brackets by the number of samples harbouring a mutation. **d** Survival analysis of pulmonary carcinoids based on the expression level of eight core genes of cluster Carcinoid B. The genes were selected using a regularised GLM on expression data. For each gene, coloured lines correspond to the Kaplan–Meier curve of overall survival for individuals with a high (green) and low (orange) expression level of this gene. Cutoffs for the two groups were determined using maximally selected rank statistics (Methods). The percentage of samples in each group is represented above each Kaplan–Meier curve and the logrank test p-value is given in bottom right for each gene. Data necessary to reproduce the figure are provided in Supplementary Data 5, 10, and in the European Genome-phenome Archive

expected, LCNECs and SCLCs carried high levels of this gene (FPKM > 1 in 99% and 92% of the samples, respectively). Although the levels of *MKI67* for each of the clusters were different, further analyses showed that *MKI67* expression levels alone were not able to accurately separate good- from poor-prognosis pulmonary carcinoids (Supplementary Fig. 11B, C).

An overview of the different molecular groups of pulmonary carcinoids and their most relevant characteristics is displayed in Fig. 6.

## Discussion

Lung neuroendocrine neoplasms are a heterogeneous group of tumours with variable clinical outcomes. Here, we characterised and contrasted their molecular profiles through integrative analysis of transcriptome and methylome data, using both machine-learning (ML) techniques and multi-omics factor analyses (MOFA). ML analyses showed that the molecular profiles could distinguish survival outcomes within patients with atypical carcinoid morphological features, splitting them into patients with good typical-carcinoid-like survival and patients with a clinical outcome similar to LCNEC. Overall, out of the 35 histopathologically atypical carcinoids, ML reclassified 12 into the typical category.

Unsupervised MOFA and subsequent gene-set enrichment analyses unveiled the immune system and the retinoid and xenobiotic metabolism as key deregulated processes in pulmonary carcinoids, and identified three molecular groups—clusters—with clinical implications (Fig. 6). The first group (cluster A1) presented high infiltration by dendritic cells, which are believed to promote the recruitment of immune effector cells resulting in a strongly active immunity[25]. Samples in cluster A1 showed overexpression of *ASCL1* and *DLL3*. The transcription factor ASCL1 is a master regulator that induces neuronal and neuroendocrine differentiation. It regulates the expression of *DLL3*, which encodes an inhibitor of the Notch pathway[26]. Overexpression of *ASCL1* and *DLL3* is a characteristic of the SCLC of the classic subtype[26] and of type-I LCNEC[12]. We validated the expression of DLL3 in an independent series of 20 pulmonary carcinoids assessed by immunohistochemistry (IHC; 40% positive). The fact that we found a correlation between the protein levels of DLL3 and CD1A (a marker of dendritic cells also assessed by IHC in this series; 60% positive) provides orthogonal evidence to support the existence of this molecular group. Phase I trials have provided evidence for clinical activity of the anti-DLL3 humanised monoclonal antibody in high–DLL3-expressing SCLCs and LCNECs[27], and additional clinical trials are ongoing in other cancer types.

The second group (cluster A2) harboured recurrent somatic mutations in *EIF1AX*, and showed downregulation of the *SLIT1*

and *ROBO1* genes. SLIT and ROBO proteins are known to be axon-guidance molecules involved in the development of the nervous system[28], but the SLIT/ROBO signalling has also been associated with cancer development, progression, and metastasis. Pulmonary neuroendocrine cells (PNEC) represent 1% of the total lung epithelial cell population[29], they reside isolated (Kultchinsky cells) or in clusters named neuroepithelial bodies (NEBs), and are believed to be the cell of origin of most lung neuroendocrine neoplasms[30]. In the normal lung, it has been shown that *ROBO1/2* are expressed, exclusively, in the PNECs, and that the SLIT/ROBO signalling is required for PNEC assembly and maintenance in NEBs[31]. In cancer, this pathway mainly suppresses tumour progression by regulating invasion, migration, and apoptosis, and therefore, is often downregulated in many cancer types[28]. More specifically, the SLIT1/ROBO1 interaction can inhibit cell invasion by inhibiting the SDF1/CXCR4 axis, and can attenuate cell cycle progression by destruction of β-catenin and CDC42[28]. Potential clinical avenues to this finding exist, especially the ongoing development of CXCR4 inhibitors.

The third molecular group (cluster B) was enriched in monocytes and depleted of dendritic cells, and had the worst median survival. Even in the presence of T cell infiltration, this immune contexture suggests an inactive immune response, dominated by monocytes and macrophages with potent immunosuppressive functions, and almost devoid of the most potent antigen-presenting cells, dendritic cells, suggesting dendritic cell-based immunotherapy as a therapeutic option for this group of samples[32]. Cluster B was also characterised by recurrent somatic mutations in *MEN1*, the most frequently altered gene in pulmonary carcinoids and pancreatic NETs[33], which is in line with the common embryologic origin of pancreas and lung. *MEN1* was inactivated by genomic rearrangement due to a chromothripsis event affecting chromosomes 11 and 20 in one of our samples. This observation, together with two additional reported cases involving chromosomes 2, 12, and 13[11], and chromosomes 2, 11, and 20[34], respectively, suggest that chromothripsis is a rare but recurrent event in pulmonary carcinoids. Interestingly, *MEN1* mutations did not have a clear prognostic value in our series. Regarding the above-mentioned deregulation of the retinoid and xenobiotic metabolism in pulmonary carcinoids, samples in cluster B presented high levels of UGT and CYP genes. In line with previous studies[35,36], these samples also harboured low levels of *OTP*, which gene expression levels were correlated with survival in the ML predictions. High levels of *ANGPTL3* and *ERBB4* were also detected in this group of samples, representing candidate therapeutic opportunities. ANGPTL3 is involved in new blood vessel growth and stimulation of the MAPK pathway[37]. This protein has been found aberrantly expressed in several types

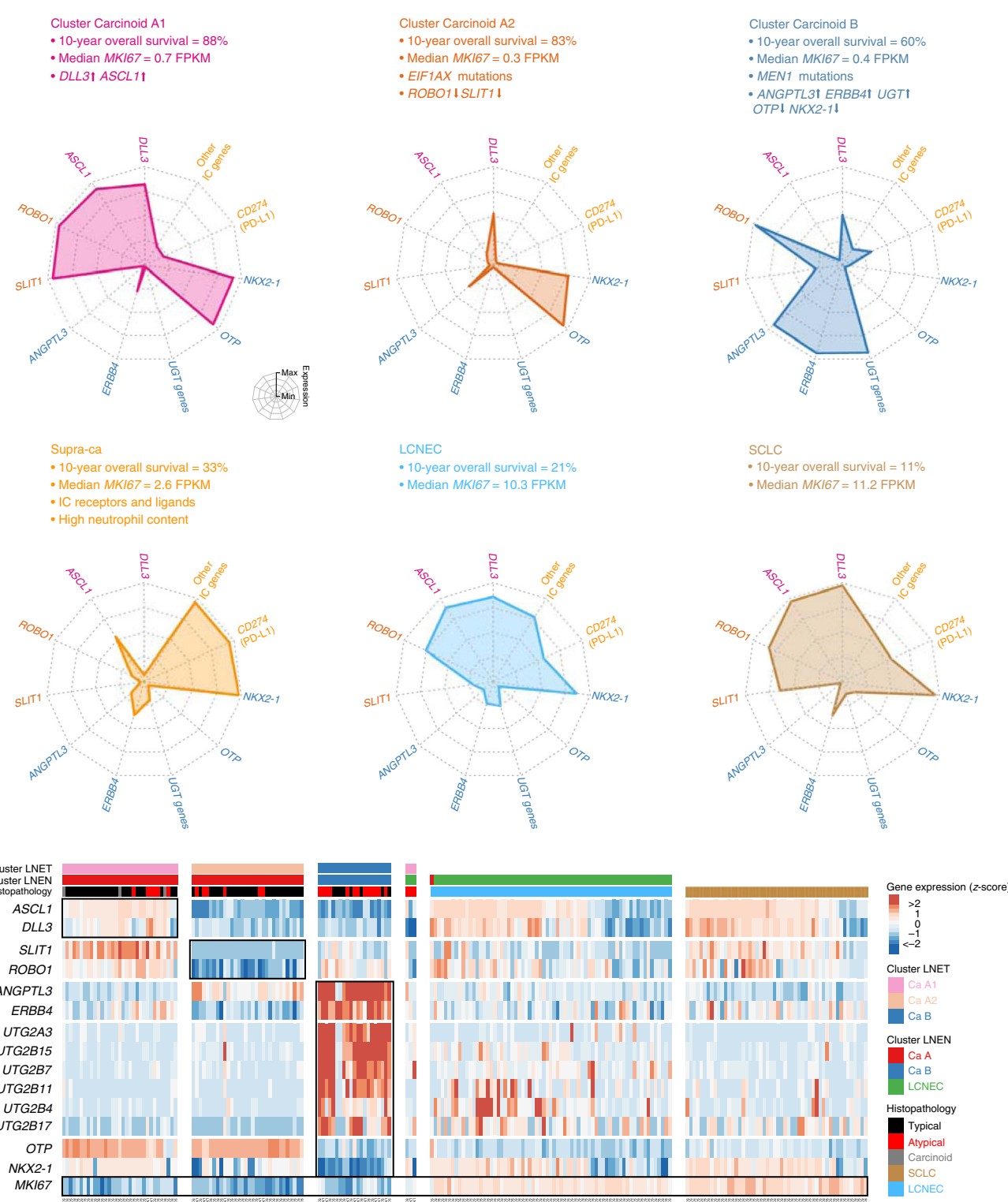

**Fig. 6** Main molecular and clinical characteristics of lung neuroendocrine neoplasms. Upper panel: Radar charts of the expression level (z-score) of the characteristic genes [*DLL3, ASCL1, ROBO1, SLIT1, ANGPTL3, ERBB4*, UGT genes family, *OTP, NKX2-1*, PD-L1 (*CD274*), and other immune checkpoint genes] of each LNET molecular cluster (Carcinoid A1, Carcinoid A2, and B clusters), supra-ca, LCNEC, and SCLC. The coloured text lists relevant characteristics—additional molecular, histopathological, and clinical data—of each group. Lower panel: heatmap of the expression level (z-score) of the characteristic genes of each group from the left panel, expressed in z-scores. Data necessary to reproduce the figure are provided in the European Genome-phenome Archive

of human cancers[37]. Similarly, overexpression of the epidermal growth factor receptor *ERBB4*, which induces a variety of cellular responses, including mitogenesis and differentiation, has also been associated with several cancer types[38,39].

For many years, it has been widely accepted that the lung well-differentiated NETs (typical and atypical carcinoids) have unique clinico-histopathological traits with no apparent causative relationship or common genetic, epidemiologic, or clinical traits with

the lung poorly differentiated SCLC and LCNEC[3]. While molecular studies have sustained this belief for pulmonary carcinoids vs. SCLC[11,13,14], the identification of a carcinoid-like group of LCNECs[10,12], the recent observation of LCNEC arising within a background of pre-existing atypical carcinoid[40], and a recent proof-of-concept study supporting the progression from pulmonary carcinoids to LCNEC and SCLC[9], suggest that the separation between pulmonary carcinoids and LCNEC might be more subtle than initially thought, at least for a subset of patients. Our study supports the suggested molecular link between pulmonary carcinoids and LCNEC, as we have identified a subgroup of atypical carcinoids, named supra-carcinoids, with a clear carcinoid morphological pattern but with molecular characteristics similar to LCNEC. In our series, the proportion of supra-carcinoids was in the order of 5.5% (six out of 110 pulmonary carcinoids with available expression/methylation data); however, considering the intermediate phenotypes observed in the MOFA LNEN, the exact proportion would need to be confirmed in larger series. We found high estimated levels of neutrophil infiltration in the supra-carcinoids. For both supra-carcinoids and LCNEC (but not SCLC), the pathways related to neutrophil chemotaxis and degranulation, were also altered. Neutrophil infiltration may act as immunosuppressive cells, for example through PD-L1 expression[41]. Indeed, the supra-carcinoids also presented levels of immune checkpoint receptors and ligands (including *PDL1* and *CTLA4*) similar—or higher—than those of LCNEC and SCLC, as well as upregulation of other immunosuppressive genes such as HLA-G, and interferon gamma that is speculated to promote cancer immune-evasion in immunosuppressive environments[42,43]. If confirmed, this would point to a therapeutic opportunity for these tumours since strategies aiming at decreasing migration of neutrophils to tumoral areas, or decreasing the amount of neutrophils have shown efficacy in preclinical models[44]. Similarly, immune checkpoint inhibitors, currently being tested in clinical trials, might also be a therapeutic option for these patients.

Overall, although preliminary, our data suggest that supra-carcinoids could be diagnosed based on a combination of morphological features (carcinoid-like morphology, useful for the differential diagnosis with LCNEC/SCLC) and the high expression of a panel of immune checkpoint (IC) genes (LCNEC/SCLC-like molecular features, useful for the differential diagnosis with other carcinoids); the levels of IC genes, such as *PD-L1*, *VISTA*, and *LAG3*, could also be used to drive the therapeutic decision for patients harbouring a tumour belonging to this subset of very aggressive carcinoids. Nevertheless, due to the very low number of supra-carcinoids identified so far ($n = 6$), follow-up studies are warranted to comprehensively characterise these tumours from pathological and molecular standpoints, to evaluate the immune cell distribution, and to establish if the diagnosis of these supra-carcinoids can be undertaken in small biopsies. Finally, the current classification only recognises the existence of grade-1 (typical) and grade-2 (atypical) well-differentiated lung NETs, while the grade-3 would only be associated with the poorly differentiated SCLC and LCNEC; however, in the pancreas, stomach and colon, the group of well-differentiated grade-3 NETs are well known and broadly recognised[45]. Whether these supra-carcinoids constitute a separate entity that may be the equivalent in the lung of the gastroenteropancreatic, well-differentiated, grade-3 NETs will require further research.

In summary, this study provides comprehensive insights into the molecular characteristics of pulmonary carcinoids, especially of the understudied atypical carcinoids. We have identified three well-characterised molecular groups of pulmonary carcinoids with different prognoses and clinical implications. Finally, the identification of supra-carcinoids further supports the already

suggested molecular link between pulmonary carcinoids and LCNEC that warrants further investigation.

## Methods

**Sample collection.** All new specimens were collected from surgically resected tumours, applying local regulations and rules at the collecting site, and including patient consent for molecular analyses as well as collection of de-identified data, with approval of the IARC Ethics Committee. These samples underwent an independent pathological review. For the typical carcinoids and LCNEC, on which methylation analyses were performed, the DNA came from the samples included in already published studies[4,11–14,35], for which the pathological review had already been done.

**Clinical data.** Collected clinical data included age (in years), sex (male or female), smoking status (never smoker, former smoker, passive smoker, and current smoker), Union for International Cancer Control/American Joint Committee on Cancer stage, professional exposure, and survival (calculated in months from surgery to last day of follow-up or death). These data were merged with that from Fernandez-Cuesta et al.[11], George et al.[12], and George et al.[14]. In order to improve the power of the statistical analyses, we regrouped some levels of variables that had few samples. Age was discretized into three categories ((15, 40], (40, 60], and (60, 90] years), Union for International Cancer Control stages were regrouped into four categories (I, II, III, IV), and smoking status was regrouped into two categories (non-smoker, that includes never smokers and passive smokers, and smoker, that includes current and former smokers). In addition, one patient (S02236) that was originally classified as male was switched to female based on its concordant whole-exome, transcriptome, and methylome data; and one patient (LNEN028) for whom no sex information was available was classified as male based on its methylation data (Supplementary Fig. 28; see details of the methods used in the DNA sequencing, expression, and methylation sections of the methods), because we had no other data type for this sample. Note that two SCLC samples from George et al.[14] displayed Y chromosome expression patterns discordant with their clinical data (S02249 and S02293; Supplementary Fig. 28B), but because we did not perform any analysis of SCLC samples that used sex information, this did not have any impact on our analyses. See Supplementary Data 1 for the clinical data associated with the samples.

We assessed the associations between clinical variables—a batch variable (sample provider), the main variable of interest (histopathological type), and important biological covariates (sex, age, smoking status, and tumour stage)—using Fisher's exact test, adjusting the p-values for multiple testing. Using samples from all histopathological types (typical and atypical carcinoids, LCNEC, and SCLC), we found that the sample provider was significantly associated with the histopathological type (Supplementary Fig. 29A). Indeed, the 20 carcinoids from one of the providers (provider 1) are all atypical carcinoids. Nevertheless, because there are also seven atypical carcinoids from a second provider and five from a third one, variables provider and histopathological type are not completely confounded and we could check for batch effects in the following molecular analysis by making sure that the molecular profiles of atypical carcinoids from provider 1 overlap with that from the two other providers. The histopathological type was significantly associated with all other variables (Supplementary Fig. 29A, B, and C).

**Pathological review.** Some of the samples included in this manuscript had already undergone a Central Pathological Review in the context of other published studies, so we used the classifications from the supplementary tables of the corresponding manuscripts[4,11,12,14,35]. For the new ones, an H&E (hematoxylin and eosin) stain from a representative FFPE block was collected for all tumours for pathological review. All tumours were classified according to the 2015 WHO classification by three independent pathologists (E.B., B.A.A., and S.L.). An H&E stain was also performed in order to assess the quality of the frozen material used for molecular analyses and to confirm that all frozen samples contained at least 70% of tumour cells.

**Immunohistochemistry.** FFPE tissue sections (3 μm thick) from 20 atypical and typical carcinoids were deparaffinized and stained with the Ventana DLL3 (SP347) assay, UltraView Universal DAB Detection Kit (Ventana Medical Systems and Amplification Kit (Ventana Medical Systems—Roche) on Ventana ULTRA auto-stainer (Ventana, Roche, Meylan, France), and with the CD1 rabbit monoclonal antibody (cl EP3622) (Ventana). The positivity of DLL3 was defined by the percentage of tumour cells exhibiting a cytoplasmic staining, whatever the intensity. The positivity of CD1A was defined by the percentage of the total surface of the tumour exhibiting a membrane staining with 1 corresponding to less than 1%, 2 to a percentage between 1 and 5%, and 3 to greater than 5%. Results are presented in Supplementary Data 9 and representative slides are shown in Fig. 4c.

**Statistical analyses.** All tests involving multiple comparisons were adjusted using the Benjamini–Hochberg procedure controlling the false discovery rate[46] using the p.adjust R function (stats package version 3.4.4). All tests were two-sided. Also, a

summary of the statistics associated with survival analyses is provided in Supplementary Data 14.

**Survival analysis**. We performed survival analysis using Cox's proportional hazard model; we assessed the significance of the hazard ratio between the reference and the other levels using Wald tests, and assessed the global significance of the model using the logrank test statistic (R package survival v. 2.41-3). Kaplan–Meier and forest plots were drawn using R package survminer (v. 0.4.2). Note that three LCNEC samples from George et al.[14] had missing survival censor information and were thus excluded from the analysis (samples S01580, S01581, and S01586).

**DNA extraction**. Samples included were extracted using the Gentra Puregene tissue kit 4g (Qiagen, Hilden, Germany), following the manufacturer's instructions. All DNA samples were quantified by the fluorometric method (Quant-iT Pico-Green dsDNA Assay, Life Technologies, CA, USA), and assessed for purity by NanoDrop (Thermo Scientific, MA, USA) 260/280 and 260/230 ratio measurements. DNA integrity of Fresh Frozen samples was checked by electrophoresis in a 1.3% agarose gel.

**RNA extraction**. Samples included were extracted using the Allprep DNA/RNA extraction kit (Qiagen, Hilden, Germany), following manufacturer's instructions. All RNA samples were treated with DNAse I for 15 min at 30 °C. RNA integrity of frozen samples was checked with Agilent 2100 Electrophoresis Bioanalyser system (Agilent Biotechnologies, Santa Clara, CA95051, United States) using RNA 6000 Nano Kit (Agilent Biotechnologies).

**Whole-genome sequencing (WGS)**. Whole-genome sequencing was performed on three fresh frozen pulmonary carcinoids and matched-blood samples by the Centre National de Recherche en Génomique Humaine (CNRGH, Institut de Biologie François Jacob, CEA, Evry, France). After a complete quality control, genomic DNA (1 μg) has been used to prepare a library for whole-genome sequencing, using the Illumina TruSeq DNA PCR-Free Library Preparation Kit (Illumina Inc., CA, USA), according to the manufacturer's instructions. After normalisation and quality control, qualified libraries have been sequenced on a HiSeqX5 platform from Illumina (Illumina Inc., CA, USA), as paired-end 150 bp reads. One lane of HiSeqX5 flow cell has been produced for each sample, in order to reach an average sequencing depth of 30x for each sample. Sequence quality parameters have been assessed throughout the sequencing run and standard bioinformatics analysis of sequencing data was based on the Illumina pipeline to generate fatsq files for each sample.

**Whole-exome sequencing (WES)**. Whole-exome sequencing was performed on 16 fresh frozen atypical carcinoids in the Cologne Centre for Genomics. Exomes were prepared by fragmenting 1 μg of DNA using sonication technology (Bioruptor, Diagenode, Liège, Belgium) followed by end repair and adapter ligation including incorporation of Illumina TruSeq index barcodes on a Biomek FX laboratory automation workstation from Beckman Coulter (Beckman Coulter, Brea, CA, USA). After size selection and quantification, pools of five libraries each were subjected to enrichment using the SeqCap EZ v2 Library kit from NimbleGen (44Mb). After validation (2200 TapeStation; Agilent Technologies, CA, USA), the pools were quantified using the KAPA Library Quantification kit (Peqlab, Erlangen, Germany) and the 7900HT Sequence Detection System (Applied Biosystems, Waltham, MA, USA), and subsequently sequenced on an Illumina HiSeq 2000 sequencing instrument using a paired-end 2 × 100 bp protocol and an allocation of one pool with 5 exomes/lane. The expected average coverage was approximately 120x after removal of duplicates (11 GB).

**Targeted sequencing**. Targeted sequencing was performed on the same 16 fresh frozen atypical carcinoids and 13 matched-normal tissue for the samples with enough DNA. Three sets of primers covering 1331 amplicons of 150–200 bp were designed with the QIAGEN GeneRead DNAseq custom V2 Builder tool on GRCh37 (gencode version 19). Target enrichment was performed using the GeneRead DNAseq Panel PCR Kit V2 (QIAGEN) following a validated in-house protocol (IARC). The multiplex PCR was performed with six separated primers pools [(1) 1 pool covering 786 amplicons, (2) 4 pools covering 498 amplicons, and (3) 1 pool covering 47 amplicons]. Per pool, 20 ng (1) or 10 ng (2 and 3) of DNA were dispensed and air-dried (only 2 and 3). Subsequently 11 μL (1) or 5 μL (2 and 3) of the PCR mix were added [containing 5.5 μL (1) or 2.5 μL (2 and 3) Primer mix pool (2x), 2.2 μL (1) or 1 μL (2 and 3) PCR Buffer (5x), 0.73 μL (1) or 0.34 μL (2 and 3) HotStar Taq DNA Polymerase (6 U/μL) and 0.57 μL (1) or 1.16 μL (2 and 3) H2O] and the DNA were amplified in a 96-well-plate as following: 15 min at 95 °C; 25 (1), 21 (2), or 23 (3) cycles of 15 s at 95 °C and 4 min at 60 °C; and 10 min at 72 °C. For each sample, amplified PCR products were pooled together, purified using 1.8x volume of SeraPure magnetic beads (prepared in-house following protocol developed by Faircloth & Glenn, Ecol. And Evol. Biology, Univ. of California, Los Angeles) (1) or NucleoMag® NGS Clean-up from Macherey-Nagel (2 and 3) and quantified by Qubit DNA high-sensitivity assay kit (Invitrogen

Corporation). One-hundred nanograms of purified PCR product (6 μL) were used for the library preparation with the NEBNext Fast DNA Library Prep Set (New England BioLabs) following an in-house validated protocol (IARC). End repair was performed [1.5 μL of NEBNext End Repair Reaction Buffer, 0.75 μL of NEBNext End Repair Enzyme Mix, and 6.75 μL of H2O] followed by ligation to specific adapters and in-house prepared individual barcodes (Eurofins MWG Operon, Germany) [4.35 μL of H2O, 2.5 μL of T4 DNA Ligase Buffer for Ion Torrent, 0.7 μL of Ion P1 adaptor (double-stranded), 0.25 μL of Bst 2.0 WarmStart DNA Polymerase, 1.5 μL of T4 DNA ligase, and 0.7 μL of in-house barcodes]. Bead purification of 1.8x was applied to clean libraries and 100 ng of adaptator ligated DNA were amplified with 15 μL of Master Mix Amplification [containing 1 μL of Primers, 12.5 μL of NEBNext High-Fidelity 2x PCR Master Mix, and 1.5 μL of H2O]. Pooling of libraries was performed equimolarly and loaded on a 2% agarose gel for electrophoresis (220 V, 40 min). Using the GeneClean™ Turbo kit (MP Biomedicals, USA) pooled DNA libraries were recovered from selected fragments of 200–300 bp in length. Libraries quality and quantity were assessed using Agilent High Sensitivity DNA kit on the Agilent 2100 Bioanalyzer on-chip electrophoreses (Agilent Technologies). Sequencing of the libraries was performed on the Ion Torrent™ Proton Sequencer (Life Technologies Corp) aiming for deep coverage (> 250x), using the Ion PI™ Hi-QT™ OT2 200 Kit and the Ion PI™ Hi-Q™ Sequencing 200 Kit with the Ion PI™ Chip Kit v3 following the manufacturer's protocols.

**DNA data processing**. WGS and WES reads mapping on reference genome GRCh37 (gencode version 19) were performed using our in-house workflow (https://github.com/IARCbioinfo/alignment-nf, revision number 9092214665). This workflow is based on the nextflow domain-specific language[47] and consists of three steps: reads mapping (software bwa version 0.7.12-r1044[48], duplicate marking (software samblaster, version 0.1.22)[49], and reads sorting (software sambamba, version 0.5.9)[50]. Reads mapping for the targeted sequencing data was performed using the Torrent Suite software version 4.4.2 on reference genome hg19. Local realignment around indels was then performed for both using software ABRA (version 0.97bLE)[51] on the regions from the bed files provided by Agilent (SeqCap_EZ_Exome_v2_probe-covered.bed) and QIAGEN, respectively, for the WES and targeted sequencing data. Consistency between sex reported in the clinical data and WES data was assessed by computing the total coverage on X and Y chromosomes (Supplementary Fig. 28A).

**Variant calling and filtering on DNA**. *WES data*: We re-performed variant calling for all typical and atypical carcinoid WES, including already published data, in order to remove the possible cofounding effect of variant calling in the subsequent molecular characterisation of carcinoids. Software Needlestack v1.1 (https://github.com/IARCbioinfo/needlestack)[52] was used to call variants. Needlestack is an ultra-sensitive multi-sample variant caller that uses the joint information from multiple samples to disentangle true variants from sequencing errors. We performed two separate multi-sample variant callings to avoid technical batch effects: (1) The 16 WES atypical carcinoids newly sequenced in this study were analysed together with 64 additional WES samples sequenced using the same protocol from another study in order to increase the accuracy of Needlestack to estimate the sequencing error rate; (2) The 15 WES LNET (ten typical and five atypical carcinoids) previously analysed (Fernandez-Cuesta et al.)[11] were reanalysed with their matched-normal. For both variant callings, we used default software parameters except for the minimum median coverage to consider a site for calling, the minimum mapping quality, and the SNV and INDEL strand bias[13] threshold (they were set to 20, 13, 4, and 10, respectively). Annotation of resulting variant calling format (VCF) files was then performed with ANNOVAR (2018Aprl16)[53] using the PopFreqAll (maximum frequency over all populations in ESP6500, 1000G, and ExAC germline databases), COSMIC v84, MCAP, REVEL, SIFT, and Polyphen (dbnsfp30a) databases.

We performed the same variant filtering after each of the two variant callings, based on several stringent criteria. First, we only retained variants that have never been observed in germline databases or present at low frequency (≤ 0.001) but already reported as somatic in the COSMIC database. Second, we only retained variants that were in coding regions and that had an impact on expressed proteins: we filtered out silent, non-damaging single nucleotide variants (based on MCAP, REVEL, SIFT, or Polyphen2 databases) and variants present in non-expressed genes (mean and median FPKM < 0.1 over all carcinoid tumours). Additionally, for calling (2), we re-assessed the somatic status of variants reported by Needlestack in light of possible contamination errors. Indeed, Needlestack is a very sensitive caller and will sometimes detect low allelic fraction variants in normal tissue that actually come from contamination by tumour cells. In such cases the variant is found in both matched samples and is reported as germline, but we still considered a variant as somatic if its allelic fraction in the normal tissue was at least five times lower than the allelic fraction observed in the tumour.

*Targeted sequencing data*: Software Needlestack was also used to call variants on targeted sequencing data from 16 atypical carcinoids and their matched-normal tissue. We performed the calling with default parameters except for the phred-scaled $q$-value and minimum median coverage to consider a site (20 and 10, respectively). These parameters were decreased compared to WES variants calling because we wanted a larger sensitivity in the validation set than in the discovery set. The annotation procedure was the same as for WES data. No other filters were used.

*Validation*: For both previously published data and data generated in this study, we only report somatic mutations that were validated using a different technique: targeted sequencing, RNA sequencing (see below for variant calling in RNA-seq data), or Sanger sequencing. Results are presented Supplementary Data 4.

**Structural variant calling**. Somatic copy number variations (CNVs) were called from WGS data using an in-house pipeline (software WGinR, available at https://github.com/aviari/wginr) that consists of three main steps. First, the dependency between GC content and raw read count is modelled using a generalised additive smoothing model with two nested windows in order to catch short and long distance dependencies. The model is computed on a subset of human genome mappable regions defined by a narrow band around the mode of binned raw counts distribution. This limits the incorporation of true biological signal (losses and gains) by selecting only regions with (supposedly) the same ploidy. In a second step, we collect heterozygous positions in the matched-normal sample and GC-corrected read counts (RC) and alleles frequencies (AF) at these positions are used to estimate the mean tumour ploidy and its contamination by normal tissue. This ploidy model is then used to infer the theoretical absolute copy number levels in the tumour sample. In the third step, a simultaneous segmentation of RC and AF signals (computed on all mappable regions) is performed using a bivariate Hidden Markov Model to generate an absolute copy number and a genotype estimate for each segment.

Somatic structural variants (SV) were identified using an in-house tool (crisscross, available at https://github.com/anso-sertier/crisscross) that uses WGS data and two complementary signals from the read alignments: (a) discordant pair mapping (wrong read orientation or incorrect insert-size) and (b) soft-clipping (unmapped first or last bases of reads) that allows resolving SV breakpoints at the base pair resolution. A cluster of discordant pairs and one or two clusters of soft-clipped reads defined an SV candidate: the discordant pairs cluster defined two associated regions, possibly on different chromosomes and the soft-clipped reads cluster(s), located in these regions, pinpointed the potential SV breakpoint positions. We further checked that the soft-clipped bases at each SV breakpoint were correctly aligned in the neighbourhood of the associated region. SV events were then classified as germline or somatic depending on their presence in the matched-normal sample. Results are presented as Supplementary Data 8 and one sample is highlighted in Fig. 3c.

**Gene-set enrichment analysis of somatic mutations**. Gene-set enrichment for somatic mutations was assessed independently for each set of Hallmark of cancer genes[18] using Fisher's exact test. We built the contingency tables used as input of the test taking into account genes with multiple mutations and used the fisher.test R function (stats package version 3.4.4). We also included validated mutations (we removed silent and intron/exon mutations) reported in SCLC[13]. In each group the p-values given by Fisher's exact test performed for all Hallmarks were adjusted for multiple testing. Supplementary Data 5 lists the altered hallmarks, including the mutated genes and the associated q-value for each group, as well as the mutated genes for each hallmarks present in each supra-carcinoid, cluster LNET, LCNEC, and SCLC samples.

We performed several robustness analyses to assess the validity of our results, in particular with regards to outlier samples/genes that would have a high leverage on the statistical results, i.e., that would alone drive the significance of a particular hallmark. First, we assessed the leverage of each individual sample using a jackknife procedure (i.e., for each sample, we performed the GSE test after removing this sample). Second, we assessed the leverage of each gene using a jackknife procedure (i.e., for each gene, we performed the GSE test without this gene). We observed that when we removed sample LNEN010 from the cluster LNET B, the sustaining proliferative signalling hallmark enrichment became non-significant at the 0.05 false discovery rate threshold, but was still significant at the 10% threshold (q-value = 0.075; Supplementary Data 3). Similarly, we observed that for several SCLC samples, once the sample was removed, the deregulating cellular energetics and inducing angiogenesis hallmarks became significant at the 0.05 false discovery rate threshold (Supplementary Data 5). For supra-carcinoids samples, we performed GSE for each sample individually. The code used for the gene set enrichment analyses on somatic mutations (Hallmarks_of_cancer_GSEA.R) is available in the Supplementary Software file 1 and the associated results are reported in Supplementary Data 5.

**RNA sequencing**. RNA sequencing was performed on 20 fresh frozen atypical carcinoids in the Cologne Centre for Genomics. Libraries were prepared using the Illumina® TruSeq® RNA sample preparation Kit. Library preparation started with 1 μg total RNA. After poly-A selection (using poly-T oligo-attached magnetic beads), mRNA was purified and fragmented using divalent cations under elevated temperature. The RNA fragments underwent reverse transcription using random primers. This is followed by second strand complementary DNA (cDNA) synthesis with DNA Polymerase I and RNase H. After end repair and A-tailing, indexing adapters were ligated. The products were then purified and amplified (14 PCR cycles) to create the final cDNA libraries. After library validation and quantification (Agilent 2100 Bioanalyzer), equimolar amounts of library were pooled. The pool was quantified by using the Peqlab KAPA Library Quantification Kit and the

Applied Biosystems 7900HT Sequence Detection System. The pool was sequenced by using an Illumina TruSeq PE Cluster Kit v3 and an Illumina TruSeq SBS Kit v3-HS on an Illumina HiSeq 2000 sequencer with a paired-end (101x7x101 cycles) protocol.

**RNA data processing**. The 210 raw reads files (89 carcinoids, 69 LCNEC, 52 SCLC) were processed in three steps using the RNA-seq processing workflow based on the nextflow language[47] and accessible at https://github.com/IARCbioinfo/RNAseq-nf (revision da7240d). (i) Reads were scanned for a part of Illumina's 13 bp adapter sequence 'AGATCGGAAGAGC' at the 3′ end using Trim Galore v0.4.2 with default parameters. (ii) Reads were mapped to reference genome GRCh37 (gencode version 19) using software STAR (v2.5.2b)[54] with recommended parameters[55]. (iii) For each sample, a raw read count table with gene-level quantification for each gene of the comprehensive gencode gene annotation file (release 19, containing 57,822 genes) was generated using script htseq-count from software htseq (v0.8.0)[56]. Gene fragments per kilobase million (FPKM) of all genes from the gencode gene annotation file were computed using software StringTie (v1.3.3b)[57] in single pass mode (no new transcript discovery), using the protocols from Pertea et al.[57] (nextflow pipeline accessible at https://github.com/IARCbioinfo/RNAseq-transcript-nf; revision c5d114e42d).

Quality control of the samples was performed at each step. Software FastQC (v. 0.11.5; https://www.bioinformatics.babraham.ac.uk/projects/fastqc/) was used to check raw reads quality, software RSeQC (v. 2.6.4) was used to check alignment quality (number of mapped reads, proportion of uniquely mapped reads). Software MultiQC (v. 0.9)[58] was used to aggregate the QC results across samples. Concordance between sex reported in the clinical data and sex chromosome gene expression patterns was performed by comparing the sum of variance-stabilised read counts (vst function from R package DESeq2) of each sample on the X and Y chromosomes (Supplementary Fig. 28B).

**Variant calling on RNA**. Software Needlestack was also used to call variants on the 20 RNA sequencing data for WES variant validation. Default parameters were used, except for the phred-scaled q-value, minimum median coverage to consider a site, and minimum mapping quality (20, 10, and 13, respectively). The annotation procedure was the same as for WES data.

**Fusion transcript detection**. RNA-seq data was processed as previously described[11,13] to detect chimeric transcripts. In brief, paired-end RNA-seq reads were mapped to the human reference genome (NCBI37/hg19) using GSNAP. Potential chimeric fusion transcripts were identified using software TRUP[59] by discordant read pairs and by individual reads mapping to distinct chromosomal locations. The sequence context of rearranged transcripts was reconstructed around the identified breakpoint and the assembled fusion transcript was then aligned to the human reference genome to determine the genes involved in the fusion. All interesting fusion-transcript were validated by Sanger sequencing. The code used for the fusion transcript detection is available on https://github.com/ruping/TRUP. All the associated results are presented Supplementary Data 7, and selected genes are highlighted in Fig. 3b.

**Unsupervised analyses of expression data**. The raw read counts of 57,822 genes from the 210 samples were normalised using the variance stabilisation transform (vst function from R package DESeq2 v1.14.1)[60]; this transformation enables comparisons between samples with different library sizes and different variances in expression across genes. We removed genes from the sex-chromosomes in order to reduce the influence of sex on the expression profiles, resulting in a matrix of gene expression with 54,851 genes and 210 samples. We performed four analyses, with different subsets of samples. (i) An analysis with all 210 samples (LNEN and SCLC), (ii) an analysis with LNEN samples only (158 samples), (iii) an analysis with LNET and SCLC samples only (139 samples), and (iv) an analysis with LNET samples only (89 samples). For each analysis, the most variable genes (explaining 50% of the total variance in variance-stabilised read counts) were selected (6398, 6009, 6234, and 5490 genes, respectively, for i, ii, iii, and iv). Principal component analysis (PCA) was then performed independently for each analysis (function dudi. pca from R package ade4 v1.7-8)[61]. Results are presented in Supplementary Fig. 6; see the Multi-omic integration section of the methods for a comparison of the results of the unsupervised analysis of expression data with that of the other 'omics.

We used the results from the PCA to detect outliers and batch effects in the expression data set. We did not detect any outliers in any of the analyses from Supplementary Fig. 6. We further studied the association between expression data, batch (sample provider), and five clinical variables of interest (histopathological type, age, sex, smoking status, and stage) using a PCA regression analysis. For each principal component, we fitted separate linear models with each of the six covariables of interest (provider plus the five clinical variables) and adjusted the resulting p-values for multiple testing. Results highlighted an association between principal component 2 and provider, histopathological type, and sex, and an association between principal components 4 and 5 and stage (Supplementary Fig. 30A). The fact that both histopathology and sample provider are jointly significantly associated with PC2 is expected given their non-independence (Supplementary Fig. 29A, B). In order to assess whether there was a batch effect

explaining the variation on PC2, we investigated the range of samples from each provider on PC2 (Supplementary Fig. 30B). We can see that samples from Provider 1 and provider 2 span a similar range on PC2 (from values less than –20 to values greater than 40). Restricting the analysis to atypical carcinoids, we can further see that AC samples from provider 2 have a range included in that of provider 1, which is expected given their differing sample sizes (five from provider 2 compared to 20 from provider 1). Overall, this shows that samples from the two providers have similar profiles and can be combined. In addition, we found that the samples that were independently sequenced in a previous study[11] and in this study (samples S00716_A and S00716_B, respectively) were spatially close in the PCA (technical replicates highlighted in Supplementary Fig. 30B).

**Supervised analysis of expression data.** We performed three distinct differential expression (DE) analyses. (i) A comparison between histopathological types; (ii) A comparison between pulmonary carcinoid (LNET) clusters A1, A2, and B (see Fig. 5a and the Multi-omic integration method section); (iii) a comparison between lung neuroendocrine neoplasm (LNEN) clusters Carcinoid A, Carcinoid B, and LCNEC (see the Multi-omic integration method section).

For each differential expression (DE) analysis, among the 57,822 genes from the raw read count tables, genes that were expressed in less than 2 samples were removed from the analysis, using a threshold of 1 fragment per million reads aligned. We also removed samples with missing data in the variables of interest (either histopathological types, LNET clusters, or LNEN clusters) or in any of the clinical covariables included in the statistical model (sex and age). This resulted in excluding two samples with missing age data from the three analyses (samples S01093, S02236), and further excluding three samples with no clear histopathological type (classified as carcinoids in Supplementary Data 1) from analysis (i) (samples S00076, S02126, S02154). For each analysis, we then identified DE genes from the raw read counts using R package DESeq2 (v. 1.21.5)[60]. For each analysis, we fitted a model with the variable of interest (type, LNET cluster, or LNEN cluster) and using sex (two levels: male and female), and age (three levels: (16, 40], (40, 60], (60, 90]) as covariables. We then extracted DE genes between each pair of groups, and adjusted the p-values for multiple testing. In order to select the genes that have the largest biological effect, we tested the null hypothesis that the two focal groups had less than 2 absolute $\log_2$-fold changes differences. For each analysis, we define the core genes of a focal group as the set of genes that are DE in all pairwise comparisons between the focal group and other groups; they correspond to genes, which expression level is specific to the focal group. For example, given three groups—A, B, and C—to find core genes, which expression levels uniquely define A compared to both B and C, we select DE genes that differentiate A from B (A vs. B), DE genes that differentiate A from C (A vs. C) and take the intersection of these gene sets [(A vs. B)∩(A vs. C)]. The code used for the DE analyses (RNAseq_supervised.R) is available at https://github.com/IARCbioinfo/RNAseq_analysis_scripts. Results of analysis (i) are reported in Supplementary Data 15 and Supplementary Data 31; results of analysis (ii) are reported in Supplementary Data 10 and Fig. 5a; results of analysis (iii) are reported in Supplementary Data 12. See section Multi-omics integration for comparisons between the analyses based on histopathological types [analysis (i)] from all 'omics perspectives.

Note that an alternative method for finding DE genes would be to compare a focal group to all the other samples together. For example, comparing group A to both groups B and C simultaneously [denoted A vs. (B and C) or A vs. the rest]. Note that this would find genes that are DE between A and the average level of expression of B and C, and thus this alternative method would have the unwanted behaviour of including the genes that are strongly DE in the comparison of A vs. B, but with similar expression levels in A and C. In order to compare the methods we used to detect core genes with this alternative method, we performed an analysis similar to analysis (ii) but comparing a focal group to all the other samples simultaneously (A vs. the rest). The comparison between our method and the alternative one is presented in Supplementary Fig. 21 and shows that our analysis provides conservative results compared to testing the focal group vs. the rest. Indeed, core DE genes reported are almost exclusively a subset of the genes found when comparing the focal group vs. the rest.

**Immune contexture deconvolution from expression data.** We quantified the proportion of cells that belong to each of ten immune cell types (B cells, macrophages M1, macrophages M2, monocytes, neutrophils, NK cells, CD4+ T cells, CD8+ T cells, CD4+ regulatory T cells, and dendritic cells) from the RNA-seq data using software quanTIseq (downloaded 23 March 2018)[62]. quanTIseq uses a rigorous RNA-seq processing pipeline to quantify the gene expression of each sample, and performs supervised expression deconvolution in a set of genes identified as informative on immune cell types, using the least squares with equality/inequality constrains (LSEI) algorithm with a reference data set containing expected expression levels for the ten immune cell types. Importantly, quanTIseq also provides estimates of the total proportion of cells in the bulk sequencing that do and do not belong to immune cells.

We tested whether immune composition differed between histopathological types, LNET clusters, LNEN clusters, and supra-carcinoids using linear permutation tests (R package lmperm, v. 2.1.0). Permutations tests are exact statistical tests that do not rely on approximations and assumptions regarding the

data distribution, and are thus well-fitted to test whether a few samples come from the same distribution as a larger group of samples. As such, they were well-fitted to handle the tests involving supra-carcinoids, for which only three samples had RNA-seq data. For each of the three analyses (histopathology, LNET clusters, and LNEN clusters), and for each pair of groups, we fitted one model per immune cell type, with the proportion of this cell type in each sample as explained variable and the cluster membership as explanatory variable. We adjusted the p-values for multiple testing. The code used for these three analyses is available on https://icbi.i-med.ac.at/software/quantiseq/doc/index.html and the associated results are presented Figs. 2f, 4b, and Supplementary Figs. 15, 19, and 32.

**EPIC 850k methylation array.** Epigenome analysis was performed on 33 typical carcinoids, 23 atypical carcinoids, and 20 LCNEC, plus 19 technical replicates. Epigenomic studies were performed at the International Agency for Research on Cancer (IARC) with the Infinium EPIC DNA methylation beadchip platform (Illumina) used for the interrogation of over 850,000 CpG sites (dinucleotides that are the main target for methylation). Each chip encompasses eight samples, so 12 chips were needed for the 95 samples. We used stratified randomisation to mitigate the batch effects, ensuring that the three histopathological types were present on every chip, while also controlling for potential confounders (the sample provider, sex, smoking status, and age of the patient); replicates were placed on different chips.

For each sample, 600 ng of purified DNA were bisulfite converted using the EZ-96 DNA Methylation-Gold$^{TM}$ kit (Zymo Research Corp., CA, USA) following the manufacturer's recommendations for Infinium assays. Three replicates included half the amount (300 ng). Then, 200 ng of bisulfite-converted DNA was used for hybridisation on Infinium Methylation EPIC beadarrays, following the manufacturer's protocol (Illumina Inc.). This array shares the Infinium HD chemistry (Illumina Inc.) and a similar laboratory protocol used to interrogate the cytosine markers with HumanMethylation450 beadchip. Chips were scanned using Illumina iScan to produce two-colour raw data files (IDAT format).

**Methylation data processing.** The resulting IDAT raw data files were pre-processed using R packages minfi (v. 1.24.0)[63] and ENmix (v. 1.14.0)[64]. We first removed unwanted technical variation in-between arrays using functional nor-malisation of the raw two-colour intensities, and computed the β-values for the 866,238 probes and 96 samples. Then, we filtered four types of probes that could confound the analyses. (i) We removed probes on the X and Y chromosomes, because we were interested in variation between tumours and treated sex as a confounder. (ii) We removed known cross-reactive probes—i.e., probes that co-hybridise to other chromosomes and thus cannot be reliably investigated. (iii) We removed probes that had failed in at least one sample, using a detection p-value threshold of 0.01, where p-values were computed with the detection P function from R package minfi, that compares the total signal (methylated + unmethylated) at each probe with the background signal level from non-negative control probes. (iv) We removed probes associated with common SNPs—that reflect underlying polymorphisms rather than methylation profiles—using a threshold minor allele frequency of 5% in database dbSNP build 137 (function dropLociWithSnps from minfi). (v) We removed probes putatively associated with rare SNPs by detecting and removing probes with multimodal β-value distributions (function nmode.mc from R package ENmix). Next, we removed duplicated samples, randomly choosing one sample per pair so as to minimise potential discrepancies, and we removed one sample that came from a metastatic tumour rather than a primary tumour. The final data set contained the β-values of 767,781 CpGs for 76 samples.

We performed quality controls of the raw data. Two-colour intensity data of internal control probes were inspected to check the quality of successive sample preparation steps (bisulfite conversion, hybridisation). We did not find outliers when comparing the methylated/unmethylated channel intensities of all samples, nor did we find samples with overall low detection p-values (the sample with the lowest mean p-value had a value of 0.001). Concordance between the sex reported in the clinical data and the methylation data was assessed using a predictor based on the median total intensity on sex-chromosomes, with a cutoff of –2 $\log_2$ estimated copy number (function getSex from minfi). Consistently with the WES and RNA-seq data, we found one sample with a mismatch between reported and inferred sex (see results in Supplementary Fig. 28C). We investigated batch effects at the raw data level using surrogate variable analysis. We used function ctrlsva from package ENmix to compute a principal component analysis of the intensity data from non-negative control probes. We retained the first ten principal components—hereafter referred to as surrogate variables—explaining >90% of the variation in control probes intensity. The ten surrogate variables were included as covariables in later supervised analyses to mitigate the impact of batch effects on the results. We checked the association of surrogate variables with batch (chip, position on the chip, and sample provider) and clinical variables (histopathological type, age, sex, smoking status) using PCA regression analysis, fitting separate linear models to each surrogate variable with each of the seven covariables of interest and adjusted the p-values for multiple testing. We show in Supplementary Fig. 33A that surrogate variables 1, 2, 3, and 10 are significantly associated with the chip (variable Sentrix id) or position on the chip (variable Sentrix position), while surrogate variables 4, 5, and 10 are significantly associated with the sample provider. The

code used to perform all the pre-processing procedure of these data is available at https://github.com/IARCbioinfo/Methylation_analysis_scripts.

**Unsupervised analysis of methylation data**. The $\beta$-values of 767,781 CpGs for 76 samples were transformed into $M$-values to perform unsupervised analyses; indeed, contrary to $\beta$-values, $M$-values theoretically range from $-\infty$ to $+\infty$ and are considered normally distributed. We performed two analyses, with different subsets of samples: (i) an analysis with all carcinoid and LCNEC samples (76 samples), and (ii) an analysis with carcinoid samples only (56 samples). For each analysis, the most variable CpGs (explaining 5% of the total variance in $M$-values) were selected (8,483 and 7,693 CpGs, respectively, for (i) and (ii). PCA was then performed independently for each analysis (function dudi.pca from R package ade4 v1.7-8)[61]. Results are presented in Supplementary Fig. 7; see the Multi-omic integration section of the methods for a comparison of the results of the unsupervised analysis of methylation data with that of the other 'omics.

We used the results from the PCA to detect outliers and batch effects in the methylation data set. We did not detect any outliers in any of the analyses from Supplementary Fig. 7. We also performed a PCA regression analysis using the same protocol as described in the data processing section above. Results highlighted no association between any principal component and array batches (chip and position in the chip; Supplementary Fig. 33A). Principal component 2 was associated with the sample provider; further examination of PCA (Supplementary Fig. 33B) revealed that this effect was driven by the samples from provider 1, which have the largest range of coordinates on PC2 (from $< -30$ to $>100$). Nevertheless, the fact that their coordinates on PC2 overlap with that of samples from other providers, and the fact that the vast majority of atypical carcinoid samples come from one provider, suggest that the large range of values of provider 1 samples on PC2 is driven by the biological variability of carcinoid methylation profiles. In addition, note that samples that cluster with LCNEC are not solely from provider 1. We assessed the impact of functional normalisation on batch effects by performing the same analysis on the $M$-values of the 5% most variable CpGs obtained without normalisation (Supplementary Fig. 33A). Compared to the PCA of the 5% most variable CpGs with normalisation (Supplementary Fig. 33A), we find that the chip position (variable Sentrix position) is significantly associated with PC10, and that PC2 is not associated with histopathology. This suggests that the functional normalisation reduced batch effects and revealed some of the biological variability in methylation data.

The PCA is also informative about associations between methylation profiles and clinical variables. We find a significant association between PC1, histopathological type, age, and smoking status, with LCNEC, smokers, and larger age classes located at higher PC1 coordinates (Supplementary Fig. 33A); these associations are expected, given that the difference between LCNEC and carcinoids is expected to be the main driver of variation in methylation, and given known the aetiology of the diseases[8]. We find an association between principal component 2, histopathology, and sex, with male and atypical carcinoids having overall larger PC2 coordinates. We find associations of larger components, in particular PC3 and age, and PC7 and 9, and sex.

**Supervised analysis of methylation data**. We detected differential methylation at the probe level (DMP) in three independent analyses: (i) between histopathological types (TC, AC, and LCNEC), (ii) between LNET clusters (clusters A1, A2, and B), and (iii) between LNEN clusters (clusters A, B, and LCNEC).

To detect DMPs, for each analysis, linear models were first fitted independently for each CpG to its $M$-values (function lmFit from R package limma version 3.34.9)[65], using the variable of interest (histopathology, LNET cluster, or LNEN cluster), in addition to the sex, age group, and the ten surrogate variables as covariates. Then, moderated $t$-tests were performed by empirical Bayes moderation of the standard errors (function eBayes from package limma), and $p$-values were computed for each CpG. Moderation enables to increase the statistical power of the test by increasing the effective degrees of freedom of the statistics, while also reducing the false-positive rate by protecting against hypervariable CpGs, and are thus favoured in array analyses. The $p$-values were adjusted for multiple testing, and CpGs with a $q$-value $<0.05$ were retained. The code used for the DMPs identification (DMP.R) is available in the Supplementary Software 1 and the associated results of analyses (i), (ii), and (iii) are presented Supplementary Data 16, Supplementary Data 11, and 17, respectively. See section Multi-omics integration for comparisons between the analyses based on histopathological types [analysis (i)] from all 'omics perspectives. Analysis (iii) confirmed most DMPs associated with DEGs reported in Fig. 5a for cluster B relative to LNET clusters (TFF1, OTOP3, SLC35D3, APOBEC2) were also DMPs for cluster B relative to LNEN clusters, showing that they harboured specific methylation levels that made them different from the LCNEC cluster, as well as from other carcinoid clusters.

**Multi-omics integration**. We performed an integrative analysis of the WES, WGS, RNA-seq, and 850 K methylation array data, using the validated somatic mutations (Supplementary Data 4), the variance-stabilised read counts, and the $M$-values, respectively. The full data set consisted of 243 samples, but some analyses focused on a subset of the data.

**Unsupervised continuous multi-omic analyses**. To perform continuous latent factors identification, we performed an integrative group factor analysis of the expression and methylation data using software MOFA (R package MOFAtools v. 0.99)[15]. MOFA identifies latent factors (LF, i.e., continuous variables) that explain most variation in the joint data sets. We did not include the somatic mutations in the model because the low level of recurrence (only few recurrently mutated genes in Supplementary Data 4) resulted in a sample by mutation matrix of much lower dimension than the other 'omics, which is known to bias the analyses[15]. Also, we did not consider expression and methylation from the sex-chromosomes, because we were interested in differences between tumours independently of the sex of the patient.

We performed four analyses, with different subsets of samples. (i) An analysis with all 235 samples for which expression or methylation data was available (LNEN and SCLC), (ii) an analysis with LNEN samples only (183 samples), (iii) an analysis with LNET and SCLC samples only (163 samples), and (iv) an analysis with LNET samples only (111 samples). For each analysis, the most variable genes for expression (explaining 50% of the total variance) were selected (6398, 6009, 6234, and 5490 genes, respectively, for i, ii, iii, and iv), and the most variable CpGs (explaining 5% of the total variance) were selected (8483, 8483, 7693, and 7693 CpGs, respectively, for i, ii, iii, and iv). Note that these lists of genes and CpGs are the same as the ones used to perform the unsupervised analyses of expression and methylation data (see above notes). Also note that we did not have EPIC 850k methylation array data for SCLC; MOFA was shown to handle missing data, including samples with entire 'omic techniques missing, by using the correlated signals from several data sets (e.g., expression and methylation) to accurately reconstruct latent factors. MOFA was performed independently for each analysis, setting the number of latent factors to 5, because subsequent latent factors explained <2% of the variance of both 'omic data sets (function runMOFA from R package MOFAtools v0.99.0). Because MOFA uses a heuristic algorithm, we assessed the robustness of the results using 20 MOFA runs. We then computed the correlations between each of the five first-latent factors across each run, resulting in a correlation matrix of 100 by 100 entries (Supplementary Figs. 2 and 17). We found that the correlations across runs were very high (> 0.95 for >80% of runs) in all analyses, suggesting that the results are robust. In addition, we found that correlations between latent factors within runs were small (typically below 0.2), which suggests that latent factors capture quasi-independent sources of variation in the data sets. For each analysis, we selected the MOFA run that resulted in the best convergence, based on the evidence lower bound statistic (ELBO). Results are presented in Figs. 1a, 4a, and Supplementary Fig. 13. Interestingly, we find that MOFA latent factors 1 to 3 for analysis (i) (LNET, LCNEC, and SCLC) correspond to MOFA LF2 to 4 for analysis (ii) (LNET and LCNEC), and to MOFA LF3 to 5 for analysis (iv) (LNET alone); this suggests that each histopathological type introduces an independent source of variation, resulting in a new LF. The code used for the unsupervised continuous molecular analyses (integration_MOFA.R) is available on https://github.com/IARCbioinfo/integration_analysis_scripts.

To perform comparisons with uni-omic unsupervised analyses, we compared the results of MOFA with that of the unsupervised analysis of expression and methylation data (Supplementary Fig. 3). To do so, we used the 51 LNEN samples for which we had both expression and methylation data, and extracted their coordinates in MOFA, expression PCA (see section unsupervised analysis of expression data), and methylation PCA (see section unsupervised analysis of methylation data). When using LNET and LCNEC samples (Supplementary Fig. 3A), we found that MOFA LF1 is strongly correlated with expression PC1 and methylation PC1 ($|r| > 0.98$; Supplementary Fig. 3D, E), and that expression PC1 and methylation PC1 are strongly correlated between them ($r = 0.97$; Supplementary Fig. 3C); LF2 was strongly correlated with expression PC3 ($r = -0.86$; Supplementary Fig. 3P), and methylation PC2 ($r = -0.98$; Supplementary Fig. 3K), suggesting that LF2 is more driven by methylation differences, but that it is nonetheless consistent with a large proportion of expression variation. On the contrary, LF3 was more strongly correlated with expression PC2 ($r = 0.87$; Supplementary Fig. 3J), suggesting that PC3 is more driven by expression differences. All these observations are consistent with the fact that the percentage of variance explained by LF2 and LF3 in terms of expression and in terms of methylation are different: LF2 explains more expression in methylation, while LF3 explains more variation in expression (Fig. 1a); it is also coherent with the fact that clusters A1 and A2 are the most separated clusters on expression PC2 (Supplementary Fig. 6B), while clusters A1 and B are the most separated on methylation PC2 (Supplementary Fig. 7A). When using LNET samples only (Supplementary Fig. 3B), we found that MOFA LF1 is strongly correlated with expression PC2 and methylation PC1 ($|r| > 0.86$; Supplementary Fig. 3M, H), and that expression PC2 and methylation PC1 are strongly correlated between them ($r = 0.72$; Supplementary Fig. 3F); LF2 was strongly correlated with expression PC1 ($r = -0.88$; Supplementary Fig. 3G), and methylation PC2 ($r = 0.90$; Supplementary Fig. 3N), suggesting that LF2 is more driven by methylation differences, but that it is nonetheless consistent with a large proportion of expression variation. Again, all these observations are consistent with the fact that the percentage of variance explained by LF1 and LF2 in terms of expression and in terms of methylation are different (Fig. 4a); it is also coherent with the fact that clusters A1 and A2 are the most separated clusters on expression PC1 (Supplementary Fig. 6D), while clusters A1 and B are the most separated on methylation PC2 (Supplementary Fig. 7B).

To perform associations of latent factors with other variables, we used the results from MOFA to detect outliers and batch effects in the data set. We did not

detect any outliers in any of the analyses from Supplementary Fig. 13. We further studied the associations between the first 5 LFs, batch (sample provider), and five clinical variables of interest (histopathological type, age, sex, smoking status, and stage) using regression analysis. For each latent factor, we fitted a linear model with the six covariables of interest (provider plus the five clinical variables). Because of the reported association between sex, age, and smoking status, we also included in the model the interaction between sex and smoking status and between age and smoking status; we adjusted the resulting $p$-values for multiple testing. Significant associations ($q$-value < 0.05) are highlighted in Figs. 1a and 4a.

We also tested the association between MOFA clusters and mutations using regression analysis. We tested genes recurrently mutated in carcinoids, using a threshold of three samples (following Argelaguet et al.)[15]; indeed, non-recurrent genes are not informative about molecular groups. Only two genes were retained: *MEN1* and *EIF1AX*. We also included recurrently mutated genes reported in LCNEC[12]. Results are highlighted in Fig. 4a. Similarly, we tested the association between pathways highlighted in Supplementary Fig. 16 (Lysine demethyltransferases, polycomb complex, SWI/SNF complex) and MOFA LF using regression analysis, but did not find any significant association at a false discovery rate threshold of 0.05.

**Unsupervised discrete multi-omic analyses.** We identified molecular clusters—groups of samples with similar molecular profiles—from MOFA results. Following Mo et al.[66], given a specified number of clusters $K$, we used the $K - 1$ latent factors that explained most of the variation to perform clustering; this choice of number of latent factors in Mo et al.[66] is said to be primarily motivated by "a general principle for separating $g$ clusters among the $n$ datapoints, a rank-$k$ approximation where $k \leq g - 1$ is sufficient." In addition, because the MOFA latent factors explaining the most variance in gene expression and methylation are expected to capture more biological signal compared to the ones explaining the least variance—expected to represent more of the noise in the data set—we expect that using the first $K - 1$ latent factors would provide more biologically meaningful clusters than using all latent factors. In addition, following the procedure from Wilkerson and Hayes[67], we performed consensus clustering to detect robust molecular clusters. This procedure involved multiple replicate clusterings ($K$-means algorithm; R function kmeans), each on latent factors from an independent MOFA run done on a sub-sample (80%) of the data. Pairwise consensus values were defined as the proportion of runs in which two samples are clustered together and used as a similarity measure, and used to perform a final hierarchical clustering (median linkage method). Consensus clustering results for $K$ from 2 to 5, for LNET plus LCNEC samples, and LNET samples alone, are presented in Supplementary Figs. 5 and 18, respectively. In the case of LNET alone, because the optimal Dunn index, which evaluates the quality of clustering as a ratio of within-cluster to between-cluster distances, corresponded to $K = 3$ clusters (Supplementary Fig. 18C), we chose the solution with three clusters. Nevertheless, note that the cluster memberships for $K = 4$ and $K = 5$ are almost perfectly nested into that for $K = 3$ (e.g., samples from the blue cluster for $K = 3$, Supplementary Fig. 18B are split between a blue and a purple cluster for $K = 4$), so the solutions with three and four clusters are coherent. Cluster memberships are highlighted in Fig. 4a. Similarly, in the case of LNET plus LCNEC samples (LNEN), because the optimal Dunn index is reached when $K = 3$, we chose that solution, but note that the cluster memberships for $K > 3$ are also nested into that for $K = 3$, so all results are coherent across values of $K$.

In order to test whether using additional latent factors could increase the power to detect molecular clusters, we performed a similar analysis but using all five latent factors identified by MOFA. In order to provide more importance to the factors most likely to capture the biological variation in the data, the multiple replicate clusterings were performed using a weighted $k$-means algorithm, where variables (here MOFA latent factors) are given weights corresponding to their proportion of variance explained. More specifically, instead of minimising the within-cluster sum of squares, the weighted within-cluster sum of squares is minimised. Results for $K = 3$ clusters of LNET and LNEN samples are presented in Supplementary Fig. 8. We can see that the alternative approach (weighted $K$-means on five latent factors) leads to the exact same cluster membership as the original approach ($K$-means on $K - 1$ latent factors), both for LNEN and LNET clusters. Indeed, among the latent factors, only the first 3 were associated with either the LNEN clusters (ANOVA $q = 4.09 \times 10^{-84}$, $8.63 \times 10^{-80}$, 0.66, 0.094, 0.24, respectively, for latent factors 1 through 5) or the LNET clusters (ANOVA $q = 5.06 \times 10^{-4}$, $5.99 \times 10^{-47}$, $5.12 \times 10^{-46}$, 0.15, 0.052, respectively), which indicates that the first three latent factors captured the differences between clusters. The code used for the clustering analyses (integration_unsupervised.R) is available at https://github.com/IARCbioinfo/integration_analysis_scripts.

**GSEA on multi-omic latent factors.** We performed gene set enrichment analysis (GSEA) on the latent factors identified by MOFA using the built-in function FeatureSetEnrichmentAnalysis[15]. This tests for each latent factor whether the distribution of the loadings of features (genes or CpGs) from a focal set are significantly different from the global distribution of loadings from features outside the set. We performed the analysis using two reference databases of gene sets: GO and KEGG. To retrieve the appropriate databases, for all genes from the muti-omics integration analysis, we downloaded GO terms using R package biomaRt[68],

and we retrieved KEGG pathways using R package KEGGgraph (v. 1.38.0)[69]. Results are presented in Supplementary Data 6.

**Expression and methylation correlation analysis.** We performed correlation tests in two analyses: (i) between LNET clusters (clusters A1, A2, and B), and (ii) between LNEN clusters (clusters A, B, and LCNEC). We selected for each gene, the set of CpGs in the region −2000 to +2000 from the transcription start site (TSS) using function getnearestTSS from R package FDb.InfiniumMethylation.hg19 version 2.2.0 based on the IlluminaHumanMethylationEPICanno.ilm10b2.hg19 annotation (get Annotation function from R package minfi version 1.24.0)[63].

We performed correlation test analyses (function cor.test from R package stats version 3.5.1) using the core genes lists (Supplementary Data 10 and 12) to find associations between expression and methylation data for each CpG, using Pearson's correlation coefficient. The $p$-values were adjusted for multiple testing. In addition, we explored the correlation between expression and methylation data by fitting a linear model independently for each correlated CpG (function lm from R package stats version 3.5.1). Finally, we calculated the interquartile distance of $\beta$-values for each CpG. CpGs with a $q$-value < 0.05, $r^2 > 0.5$ and an interquartile distance greater than 0.25 were retained and, among these CpGs, only the one with the smallest $q$-value has been represented in Supplementary Fig. 22. Results of analyses (i) and (ii) are reported in Supplementary Data 10 and 12.

**Survival analysis using penalised generalised linear model.** We computed a generalised linear model with elastic net regularisation (R package glmnet v2.0-16)[70] to select the genes associated with the survival of LNET samples. We fixed the elastic net mixing parameter $\alpha$ to 0.5 and used leave-one-out cross-validation to determine the regularisation parameter $\lambda$ (cv.glmnet function from glmnet package). To be more stringent, the optimal regularisation parameter chosen was the one associated with the most regularised model with cross-validation error within one standard deviation of the minimum. In order to identify the genes associated with the poor survival of the cluster Carcinoid B, we included in the model only the expression of the core genes of this cluster defined in the MOFA considering only the LNET samples (see section Multi-omics integration). We used the normalised read counts, and centred and scaled them using R package caret (v6.0-80). The genes with non-zero estimated coefficients are listed in Supplementary Data 13. For each non-coding gene, we determined the optimal cutpoint of expression (normalised read counts) that best separates the survival outcome into two groups using the surv_cutpoint function based on the maximally selected rank statistics and available in the R package survminer (v0.4.3). The minimal proportion of samples per group was set to 10%.

**Supervised multi-omic analyses.** We performed supervised learning in order to classify typical and atypical carcinoids, and LCNEC based on the different 'omics data available: expression and methylation data.

*Classification algorithm*: Each classification was performed using a random forest algorithm (R package randomForest v4.6-14). Considering the restricted number of samples, we performed a leave-one-out cross-validation. For each run, to increase the training set size, minority classes were oversampled so that all classes reach the same number of training samples. Note that for the sample with technical replication of RNA-seq data (S00716_A and S00716_B), in order to avoid model overfitting, the two replicates were never simultaneously included in the training and test sets. Also in order to avoid overfitting, we performed normalisation and independent feature filtering within each fold, so that test samples were excluded from this step. More specifically, for the expression data, the features of the training set were first normalised using the variance stabilisation transformation (vst function from R package DESeq2 v1.22.2), then mean-centred and scaled to unit variance. Then, the variance stabilising transformation learned from the training set was applied to the test set using the dispersionFunction function from the DESeq2 package, and centreing and scaling were performed using the values learned from the training set. For the methylation data, the $M$ values were computed using the R package minfi (v1.28.3); the features of the training set were mean-centred and scaled to unit variance, then the test sample features were centred and scaled using the values learned from the training set. For each fold of the leave-one out, the training set was used for the feature selection. Based on the training set, we selected the most variable features, representing 50% and 5% of the total variation in expression and methylation data, respectively. The code used for the machine learning analyses (ML_functions.r) is available in the Supplementary Software 1 and the associated results are reported in Supplementary Data 1.

*Defining an Unclassified category*: The random forest algorithm provides for each predicted sample the class probabilities. We considered a sample as unclassifiable (Unclassified category) if the ratio of the two highest probabilities was below 1.5. In fact, this threshold allowed us to identify a category of samples with intermediate molecular profiles, for which the algorithm assigns similar probabilities to the two most probable classes. Because of the small sample size, this parameter was chosen a priori and not tuned in order to avoid overfitting. In Supplementary Fig. 10, we compared the classification results when considering three different thresholds: 1 (which corresponds to no ratio and results in few unclassified samples, i.e., only discordant expression and methylation-based

predictions, see Integration of expression and methylation data below), 1.5 (which corresponds to the ratio reported in the main text), and 3 (which corresponds to a very stringent ratio resulting in more unclassified samples). Except for the size of the unclassified classes that depends on the ratio used, the confusion matrices for the three ratios were qualitatively similar, with most LCNEC samples correctly classified, a majority of typical correctly classified, and almost as many atypical classified as typical and classified as atypical. In addition, the survival analyses of the three models also led to similar conclusions, with atypical carcinoids classified as atypical by the machine learning having a survival that is not statistically significantly different from that of LCNEC samples but that is lower from both that of typical carcinoids predicted as typical carcinoids, and that of atypical predicted as typical. However, in the case of the largest ratio, the small number of atypical samples predicted in those categories did not enable the identification of two groups of atypical carcinoids with significant different overall survival ($p = 0.086$).

*Number of samples and features*: To classify LCNEC against atypical and typical carcinoids, 157 and 76 samples were considered using the expression and methylation data, respectively. The number of features selected in each fold of the leave-one-out are of the order of 6000 and 8000 for expression and methylation features, respectively. For the analysis based on *MKI67* only (Supplementary Fig. 31C, left panel), the only feature considered was the expression of *MKI67*.

*Integration of expression and methylation data*: As the random forest algorithm does not handle missing data directly, and because only 51 out of 182 LNEN samples had both expression and methylation data available (Supplementary Fig. 1), we performed random forest classification on expression and methylation separately, and merged the classification results by combining the two sets of ML predictions. Thus, the samples with both expression and methylation data were associated with two predictions. When the two predictions were discordant we applied the following rules: (i) if one prediction was Unclassified (see Defining an Unclassified category above) and the other a histopathological category, we chose the histopathological category (ii) if the two predictions were different histopathological categories, we chose the Unclassified category.

Note that fitting independent random forest models on each data set separately corresponds to maximising the number of samples ($n$) per model at the expense of the number of features ($p$), because each model relies only on the number of features in a single data set. An alternative approach is to maximise the number of features ($p$) by combining both data sets, at the expense of the number of samples $n$, because of the limited number of samples with both data types available. Indeed, for fixed $n$ increasing $p$ requires less parameters and leads to a higher statistical power. Nevertheless, in our case, because of missing data, increasing $p$ by using both omics layers would drastically reduce $n$, restricting our sample set ($n = 157$ and $n = 76$ for expression and methylation, respectively) to the set of samples with both layers ($n = 51$, including only a single supra-carcinoid). Given the existence of very rare entities such as the supra-carcinoids, accurately capturing the diversity of molecular profiles in the training set was our priority, and thus we chose to maximise $n$. In addition, by maximising $n$, we hypothetically ensured that we would also maximise the power of the subsequent analyses based on the ML results. To confirm this hypothesis, we performed the ML analyses on the restricted set of samples, including both expression and methylation data in the same model and compared the predictions of this model to the combined predictions based on expression and methylation data separately. We found that the predictions (confusion matrix in Supplementary Fig. 9) were similar, with 43/51 samples with both data types predicted similarly in the two models. In addition, our main finding —the existence of two groups of atypical samples, which tended to have a good and bad prognosis (red and pink curves Fig. 1b)—still held, but that limited number of samples impeded the statistical analyses. In fact, none of the Cox regression tests were significant even for the groups displaying the largest differences (e.g., ML-predicted LCNEC vs. ML-predicted typical samples), and even when comparing the histological types reported by the pathologists (bottom panel Supplementary Fig. 9). This supports our hypothesis that maximising $p$ at the expense of $n$ leads to a decrease in power in subsequent analyses due to a smaller sample size, and comforts our initial choice.

As matrix factorisation methods such as MOFA and PCA remove correlations between features by finding latent factors that summarise them, they could presumably improve the performance of ML. Nevertheless, by providing low-dimensional approximations of the data, such techniques induce a loss of information, which could reduce the performance of the ML. To assess the balance between these beneficial and detrimental effects, we also performed ML using the MOFA factors or the principal components of the PCA analysis, using factors or components that explained at least 2% of the variance (five MOFA latent factors, six expression PCs, and five methylation PCs, respectively). These analyses are presented in Supplementary Fig. 12 and led to similar classification to the results presented in the main text Fig. 1. In addition, in the case of MOFA factors, in accordance with Fig. 1, atypical carcinoids were stratified into a group with an overall survival similar to that of the LCNEC (in red) and a group with a higher overall survival (in pink), similar to that of the typical carcinoids. When using the principal components, despite a similar trend, the difference in survival between the high- and low-survival groups was not significant. These results show that dimensionality reduction does not lead to an increased classification ability, nor does it provide a better explanation of clinical behaviour. We thus chose to represent only the results of the ML analyses based on expression and methylation data in the main text and figures.

**Survival analysis based on expression and methylation data**. We divided the samples into different groups based on the ML predictions. We represented the Kaplan–Meier curves of the predictions groups by selecting the groups with >10 samples and gathering the unclassified samples in the same group. Using Cox's proportional hazard model and using the logrank test statistic (R package survival v2.42-3) we compared the overall survival of LCNEC, atypical and typical samples based on the histopathological classification and based on the ML predictions (Supplementary Fig. 11A). Forest plots were drawn using R package survminer (v0.4.3). The same survival analysis was performed using the ML predictions based on *MKI67* expression only (Supplementary Fig. 11C).

**Comparison between the supervised analyses of typical and atypical carcinoids**. We contrasted the results of the different supervised analyses between typical and atypical carcinoids based on clinical data, specific markers (Ki67), machine learning, differential expression, and differential methylation (Supplementary Fig. 31). Survival analyses showed a significant difference between histopathological types (Supplementary Fig. 31A). Nevertheless, the machine learning classifier based on the genome-wide expression or methylation data could not properly distinguish atypical and typical carcinoids (Supplementary Fig. 31B): there were 64–83% correctly classified typical carcinoids and only 30–41% correctly classified atypical carcinoids. The differential expression analysis showed that atypical carcinoids also presented very few core differentially expressed genes (Supplementary Fig. 31C, middle panel and Supplementary Data 15) and differentially methylated positions (Supplementary Fig. 31C, right panel and Supplementary Data 17). Overall, these data suggest that the histopathological classification, although clinically meaningful, does not completely match the molecular classification.

**Reporting summary**. Further information on research design is available in the Nature Research Reporting Summary linked to this article.

## Data availability

The exome sequencing data, RNA-seq data, and methylation data have been deposited in the European Genome-phenome Archive (EGA) database, which is hosted at the EBI and the CRG, under accession number EGAS00001003699. Other data sets referenced during the study are available from the EGA website under accession numbers EGAS00001000650 (pulmonary carcinoids)[11], EGAS00001000708 (LCNEC)[12], and EGAS00001000925 (SCLC)[13,14]. All the other data supporting the findings of this study are available within the article and its supplementary information files and from the corresponding author upon reasonable request. A reporting summary for this article is available as a Supplementary Information file.

## Code availability

The code and software sources from previously published algorithms used to perform the analyses are detailed in the supplementary tables and online methods. Custom scripts are provided in the Supplementary Software 1. All sources for the software used in the manuscript are summarised in Supplementary Data 18.

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

## Acknowledgements

We thank the patients donating their tumour specimens. We also thank Prof. Roman K. Thomas, Dr. Martin Peifer, Dr. Julie George, Dr. Paul Brennan, and Dr. Ghislaine Scelo for their help with logistics. We also thank Dr. Ricard Argelaguet for his advice in using MOFA. This study is part of the lungNENomics project and the Rare Cancers Genomics initiative (www.rarecancersgenomics.com). This work has been funded by the US National Institutes of Health (NIH R03CA195253 to L.F.C. and J.D.M.), the French National Cancer Institute (INCa, PRT-K-17-047 to L.F.C. and TABAC 17-022 to J.D.M.), the Ligue Nationale contre le Cancer (LNCC 2016 to L.F.C.), France Genomique (to J.D.M.), and the Italian Association for Cancer Research (AIRC) (IG 19238 to M.V. and MFAG 12983 to L.A.M.) (Special Programme 5X1000, ED No12162 to U.P., L.R., and G.S.). J.S. is a Miguel Servet researcher (CP13/00055 and PI16/0295). L.M. and T.M.D. have fellowships from the LNCC.

## Author contributions

L.F.C. conceived and designed the study. L.F.C. and M.F. supervised all the aspects of the study. A.G., A.B., J.A., F.L.C.K., S.B., J.S., N.G. and S.Lan. supervised some aspects of the study. B.A.A., E.B. and S.Lan. performed the histopathological review. N.Leb., T.G., J.D., A.C., C. Cu., G.D. and N.Lem. did the lab work. N.A., N.Leb., A.A.G.G., L.M., D.H., A.S.S., A.F., T.M.D., R.O., V.M., C.V. and L.A.M. performed the computational and statistical analyses. P.L., A.C.T., A.S., J.H.C., J. Saenger, J. Stojsic, J.K.F., M.B., C.B.F., F.G.S., N.L.S., P.A.R., G.W., L.R., G.S., U.P., M.M., S.Lac., J.M.V., V.H., P.H., O.T.B., M.L.-I., V.T.M., L.A.M., P.G., M.V., M.G.P., L.B., H.P., A.M.C.D., E.B., E.J.M.S., N.G. and S.Lan contributed with samples and the corresponding histopathological, epidemiological, and clinical data. J.F.D., Z.H., A.V., P.N. and J.D.M. helped with logistics. J.D., B.A. A., C. Ca., L.R., M.M., M.V., M.G.P., L.B., H.P., G.P., J.D.M., H.H.V., E.J.M.S., N.G. and S.Lan gave scientific input. N.A., N.Leb., A.A.G.G., L.M., J.D.M., M.F. and L.F.C. wrote the manuscript, which was reviewed and commented by all the co-authors.

## Additional information

**Competing interests:** The authors declare no competing interests. Where authors are identified as personnel of the International Agency for Research on Cancer/World Health Organisation, the authors alone are responsible for the views expressed in this article and they do not necessarily represent the decisions, policy or views of the International Agency for Research on Cancer/World Health Organisation.

N. Alcala[1,35], N. Leblay[1,35], A.A.G. Gabriel[1,35], L. Mangiante[1], D. Hervas[2], T. Giffon[1], A.S. Sertier[3], A. Ferrari[3], J. Derks[4], A. Ghantous[5], T.M. Delhomme[1], A. Chabrier[1], C. Cuenin[5], B. Abedi-Ardekani[1], A. Boland[6], R. Olaso[6], V. Meyer[6], J. Altmuller[7], F. Le Calvez-Kelm[1], G. Durand[1], C. Voegele[1], S. Boyault[8], L. Moonen[4], N. Lemaitre[9], P. Lorimier[9], A.C. Toffart[10], A. Soltermann[11], J.H. Clement[12], J. Saenger[13], J.K. Field[14], M. Brevet[15], C. Blanc-Fournier[16], F. Galateau-Salle[17], N. Le Stang[17], P.A. Russell[18], G. Wright[18], G. Sozzi[19], U. Pastorino[19], S. Lacomme[20], J.M. Vignaud[20], V. Hofman[21], P. Hofman[21], O.T. Brustugun[22,23], M. Lund-Iversen[23], V. Thomas de Montpreville[24], L.A. Muscarella[25], P. Graziano[25], H. Popper[26], J. Stojsic[27], J.F. Deleuze[6], Z. Herceg[5], A. Viari[3], P. Nuernberg[7,28], G. Pelosi[29], A.M.C. Dingemans[4], M. Milione[19], L. Roz[19], L. Brcic[26], M. Volante[30], M.G. Papotti[30], C. Caux[31], J. Sandoval[2], H. Hernandez-Vargas[32], E. Brambilla[9], E.J.M. Speel[4], N. Girard[33,34], S. Lantuejoul[3,8,17], J.D. McKay[1], M. Foll[1] & L. Fernandez-Cuesta[1]

[1]International Agency for Research on Cancer (IARC/WHO), Section of Genetics, 150 Cours Albert Thomas, 69008 Lyon, France. [2]Health Research Institute La Fe, Avenida Fernando Abril Martorell, Torre 106 A 7planta, 46026 Valencia, Spain. [3]Synergie Lyon Cancer, Centre Léon Bérard, 28 Rue Laennec, 69008 Lyon, France. [4]Maastricht University Medical Centre (MUMC), GROW School for Oncology and Developmental Biology, P.O. Box 5800, 6202 AZMaastricht, The Netherlands. [5]International Agency for Research on Cancer (IARC/WHO), Section of Mechanisms of Carcinogenesis, 150 Cours Albert Thomas, 69008 Lyon, France. [6]Centre National de Recherche en Génomique Humaine (CNRGH), Institut de Biologie François Jacob, CEA, Université Paris-Saclay, 2 rue Gaston Crémieux, CP 5706, 91057 Evry Cedex, France. [7]Cologne Centre for Genomics (CCG) and Centre for Molecular Medicine Cologne (CMMC), University of Cologne, Weyertal 115, 50931 Cologne, Germany. [8]Translational Research and Innovation Department, Cancer Genomic Platform, 28 Rue Laennec, 69008 Lyon, France. [9]Institute for Advanced Biosciences, Site Santé, Allée des Alpes, 38700 La Tronche, Grenoble, France. [10]Pulmonology—Physiology Unit, Grenoble Alpes University Hospital, 38700 La Tronche, France. [11]Institute of Pathology and Molecular Pathology, University Hospital Zurich, Schmelzbergstrasse 12, 8091 Zurich, Switzerland. [12]Department Hematology and Medical Oncology, Jena University Hospital, Am Klinikum 1, 07747 Jena, Germany. [13]Bad Berka Institute of Pathology, Robert-Koch-Allee 9, 99438 Bad Berka, Germany. [14]Roy Castle Lung Cancer Research Programme, Department of Molecular and Clinical Cancer Medicine, University of Liverpool, 6 West Derby Street, L7 8TX Liverpool, UK. [15]Pathology Institute, Hospices Civils de Lyon, University Claude Bernard Lyon 1, 59 Boulevard Pinel, 69677 BRON Cedex, France. [16]CLCC François Baclesse, 3 avenue du Général Harris, 14076

Caen Cedex 5, France. [17]Department of Pathology, Centre Léon Bérard, 28, rue Laennec, 69373 Lyon Cedex 8, France. [18]St. Vincent's Hospital and University of Melbourne, Victoria Parade, Fitzroy, Melbourne, VIC 3065, Australia. [19]Pathology Division Fondazione, IRCCS Istituto Nazionale dei Tumori, Via G. Venezian 1, 20133 Milan, Italy. [20]Nancy Regional University Hospital, CHRU, CRB BB-0033-00035, INSERM U1256, 29 Avenue du Maréchal de Lattre de Tassigny, 54035 Nancy Cedex, France. [21]Laboratory of Clinical and Experimental Pathology, FHU OncoAge, Nice Hospital, Biobank BB-0033-00025, IRCAN Inserm U1081 CNRS 7284, University Côte d'Azur, 30 avenue de la voie Romaine, CS, 51069-06001 Nice Cedex 1, France. [22]Drammen Hospital, Vestre Viken Health Trust, Vestre Viken HF, Postboks 800, 3004 Drammen, Norway. [23]Institute of Cancer Research, Oslo University Hospital, Ullernchausseen 70, 0379 Oslo, Norway. [24]Marie Lannelongue Hospital, 133 avenue de la Resistance, 92350 Le Plessis Robinson, France. [25]Fondazione IRCCS Casa Sollievo della Sofferenza, Viale Cappuccini 1, 71013 San Giovanni Rotondo FG, Italy. [26]Diagnostic and Research Institute of Pathology, Medical University of Graz, Neue Stiftingtalstrasse 6, 8010 Graz, Austria. [27]Department of Thoracopulmonary Pathology, Service of Pathology, Clinical Center of Serbia, Pasterova 2, Belgrade 11000, Serbia. [28]Cologne Excellence Cluster on Cellular Stress Responses in Aging-Associated Diseases (CECAD), University of Cologne, Joseph-Stelzmann-Straße 26, 50931 Cologne, Germany. [29]Department of Oncology and Hemato-Oncology, University of Milan, and Inter-Hospital Pathology Division, IRCCS Multimedica, Via Gaudenzio Fantoli, 16/15, 20138 Milan, Italy. [30]Department of Oncology, University of Turin, Pathology Division, Via Santena 7, 10126 Torino, Italy. [31]Department of Immunity, Virus, and Inflammation, Cancer Research Centre of Lyon (CRCL), 28 Rue Laennec, 69008 Lyon, France. [32]Cancer Research Centre of Lyon (CRCL), Inserm U 1052, CNRS UMR 5286, Centre Léon Bérard, Université de Lyon, 28 Rue Laennec, 69008 Lyon, France. [33]Institut Curie, 26 Rue d'Ulm, 75005 Paris, France. [34]European Reference Network (ERN-EURACAN), 28 rue Laennec, 69008 Lyon, France. [35]These authors contributed equally: Alcala N., Leblay N., Gabriel A. A. G. [36]These authors jointly supervised this work: Foll M., Fernandez-Cuesta L.

