## [Peer Review File · Nature Communications]

Reviewers' comments:

Reviewer #1 (Remarks to the Author):

The authors have performed a detailed multi-analysis of carcinoids and LCNEC and found that atypical could be divided into two groups, one with good and the other with poor prognosis. The finding will be a basis for better diagnosis and therapy of atypical carcinoids. The machine learning methods/tools will be useful for researchers in the fields of other types of cancers. The reviewer thinks that the following points should be revised or clarified.

1. This paper uses several data analysis methods including machine learning. The authors should clarify that all the methods and source codes are open to the readers using a summary table.
2. Finding of supra-carcinoids is important. Can the authors find a common pathological features, including immune cell distribution, after knowing which are supra-carcinoids based on molecular profiling in this study? Detailed molecular profiling is necessary to discriminate supra-carcinoids from others? Is it possible for small biopsies?
3. This paper described detailed molecular characteristics of carcinoids, however, the following points are difficult to understand. 1) What molecules can distinguish/diagnose supra-carcinoids? 2) What molecules can be therapeutic targets for supra-carcinoids. Please clarify.
4. Abstract: relationship with well differentiated grade-3 gastroenteropancreatic tumors is too preliminary to be described here.
5. Main text sentences explaining Fig.1C: colours of graphs for good and poor prognosis groups must be described to help readers understand.

Reviewer #2 (Remarks to the Author):

The authors present an interesting multi-omics study, in which they identify and characterise subpopulations (clusters) of pulmonary carcinoids. While overall the extensive analyses are thoroughly conducted, I have some concerns regarding overfitting in the ML analysis and some other minor statistical concerns.

Generally, I didn't find the description of the ML analysis in the main text very clear, I suggest moving some detail in the survival analysis and the random forest fitting from the supplementary methods to the main paper. More importantly, I have some concerns regarding overfitting: the test data must not be used for preprocessing and feature selection since this „leaking“ will result in overconfident evaluations. In particular, selection of variable features as well as centering and scaling should be performed on training data only. Test data should then be normalised with mean and sd from the training data and, and features identified on the training data only should be used. Also, the 1.5 ratio for unknown is quite arbitrary. How robust are the findings wrt to this threshold?

Furthermore:

- * Why did the authors not fit a single RF model, using both methylation and expression as input, instead of having two independent models? Having two models does not leverage the interdependence of the omics layers, requires more parameters and has a lower statistical power
- * How does a ML model perform when the MOFA factors are used as inputs? How when PCs are used?
- * Please do not use the term "almost significant", this is non-sensical, statistically speaking
- * When identifying core DE genes: why not simply test group of interest vs rest? Alleviates issues regarding multiple testing in other setting
- * For the clustering, why not use all MOFA factors. I am not convinced by the citation of the iCluster+ paper as motivation for using 2 factors only: this makes sense in the way the iCluster+ framework is designed in the first place for clustering, but not for MOFA
- * It's not quite clear from the text how many factors MOFA LNEC identified, only 3? Or were only the first 3 related to clusters? If more, what about the other ones?

Reviewer 1

The authors have performed a detailed multi-analysis of carcinoids and LCNEC and found that atypical could be divided into two groups, one with good and the other with poor prognosis. The finding will be a basis for better diagnosis and therapy of atypical carcinoids. The machine learning methods/tools will be useful for researchers in the fields of other types of cancers. The reviewer thinks that the following points should be revised or clarified.

1. This paper uses several data analysis methods including machine learning. The authors should clarify that all the methods and source codes are open to the readers using a summary table.

We share the reviewer's views on the sharing of source codes to ensure results reproducibility. We now provide in **Supplementary Table S17** a link to all source codes used. Of note, all our bioinformatics workflows (e.g., alignment, quality control, variant calling) and our main routine statistical analyses scripts (e.g., methylation array processing, differential expression analysis, multi-omic factor analysis and clustering) are freely available online at <https://github.com/IARCbioinfo> under open source licenses, and the revision numbers allowing to run the exact versions of each script are provided in the **Supplementary Methods**. In addition, we provide the code for the machine learning analyses performed in this paper in a new **Supplementary File S1**, and we provide raw data for all figures in the related Supplementary Tables. Finally, note that we are in the process of submission of the data descriptors to *Scientific Data*, so that additional detailed information of the data processing and quality control steps are available in open-access.

2. Finding of supra-carcinoids is important. Can the authors find common pathological features, including immune cell distribution, after knowing which are supra-carcinoids based on molecular profiling in this study? Detailed molecular profiling is necessary to discriminate supra-carcinoids from others? Is it possible for small biopsies?

3. This paper described detailed molecular characteristics of carcinoids, however, the following points are difficult to understand. 1) What molecules can distinguish/diagnose supra-carcinoids? 2) What molecules can be therapeutic targets for supra-carcinoids. Please clarify.

We thank the reviewer for these comments that made us realize that this important message required better emphasis. We now better point out the pathological and molecular characteristics of the supra-carcinoids in **Table 1**, and provide suggestions in the discussion, about their combined use for diagnosis (**page 12**). First, we mention that supra-carcinoids can be distinguished from the high-grade lung neuroendocrine neoplasms (small-cell lung cancer and large-cell neuroendocrine carcinomas) using classical histopathological reviews, because the supra-carcinoids have in common the carcinoid cell morphology, as illustrated in **Fig. 2C**. Second, we now mention that our results suggest that supra-carcinoids could potentially be distinguished from other carcinoids using a very simple molecular profiling based on the few markers presented in **Fig. 2E** (e.g., PD-L1). We now mention in the discussion (**page 12**) that these few markers correspond to immune checkpoint genes, which suggests that this subset of very aggressive carcinoids constitutes a potential candidate for immunotherapy. Also, **Fig. 6** was modified to clarify the message, making the genes differentially expressed in supra-carcinoids more easily distinguishable. Nevertheless, we caution the readers that due to the very low number of supra-carcinoids identified so far (n=6), follow-up studies are warranted to comprehensively characterize supra-carcinoids from pathological and molecular standpoints. Similarly, this low numbers did not allowed for a meaningful immune cell distribution evaluation. In addition, since we have not addressed the intra-tumour molecular heterogeneity of these tumours, we cannot conclude if the diagnosis of supra-carcinoids based on the above-mentioned molecular characteristics could be undertaken in small biopsies.

4. Abstract: relationship with well-differentiated grade-3 gastroenteropancreatic tumors is too

We agree and have removed this sentence from the abstract.

5. Main text sentences explaining Fig.1C: colours of graphs for good and poor prognosis groups must be described to help readers understand.

We have made the suggested changes and improved the clarity of the descriptions of **Fig. 1** in **page 5** of the **main text**.

Reviewer 2

The authors present an interesting multi-omics study, in which they identify and characterise subpopulations (clusters) of pulmonary carcinoids. While overall the extensive analyses are thoroughly conducted, I have some concerns regarding overfitting in the ML analysis and some other minor statistical concerns.

1. Generally, I didn't find the description of the ML analysis in the main text very clear, I suggest moving some detail in the survival analysis and the random forest fitting from the supplementary methods to the main paper. More importantly, I have some concerns regarding overfitting: the test data must not be used for preprocessing and feature selection since this „leaking“ will result in overconfident evaluations. In particular, selection of variable features as well as centering and scaling should be performed on training data only. Test data should then be normalised with mean and sd from the training data and, and features identified on the training data only should be used.

We thank the reviewer for this helpful comment. We have now expanded the description of the machine-learning (ML) analysis in the **main text (page 4)** to include more details about both the survival model fitting and the random forest algorithm. In addition, as suggested by the reviewer, we have redone entirely the ML analyses with the test data excluded from the pre-processing (centering and scaling) and feature selection processes (briefly described in the **main text (pages 4)** and detailed in the **Supplementary Methods (pages 23 and 24)**, and have adapted **Figs. 1A-C, 4A, 6, S11, S27, and S31** accordingly. Note that in addition to the centering and scaling steps mentioned by the reviewer, in the case of the expression data, we have also excluded the test data from the variance stabilization transformation computation to avoid overfitting.

2. Also, the 1.5 ratio for unknown is quite arbitrary. How robust are the findings wrt to this threshold?

We agree with the reviewer that the 1.5 ratio is an arbitrary choice. We now provide the rationale for our choice (see **page 4** of the **main text**, and **page 24** of the **Supplementary Methods**), and also show that our results are robust to this choice of ratio in a new **Fig. S10**. Regarding this choice of ratio, we now first mention that we aimed to identify a category of samples with intermediate molecular profiles, for which the probabilities of belonging to the two main classes is close—i.e., that have a ratio close to 1. Second, we now explicitly mention that this ratio parameter was not tuned, but chosen *a priori*. Indeed, we did not initially test multiple values to choose the value leading to the best performance because of the small samples size, which did not allow tuning such a parameter without over-fitting the model and thus artificially inflating the classification performance.

Regarding the robustness of the results to the choice of ratio, we now show that our findings—the model's performance in predicting the histopathological type and the model's ability to stratify carcinoids into good- and bad-prognosis groups—are robust to this choice. We compared the results of the ML using three different ratios: 1 (which corresponds to no ratio and results in few unclassified samples), 1.5 (which corresponds to the ratio reported in the main text), and 3 (which corresponds to a very stringent ratio resulting in more unclassified samples) (**Fig. S10**). Indeed, except for the unclassified classes that depend on the ratio used,

most LCNEC samples correctly classified, a majority of typical correctly classified, and almost as many atypical classified as typical and classified as atypical. In addition, the survival analyses of the three models also led to similar conclusions, with atypical carcinoids classified as atypical by the machine learning (red curve) having a survival that is not statistically significantly different (Wald test) from that of LCNEC samples (blue curve) but that is lower than both that of typical carcinoids predicted as typical carcinoids (black curve), and that of atypical predicted as typical (pink curve). However, in the case of the largest ratio, the small number of atypical samples predicted in those categories did not allow us to identify two groups of atypical carcinoids with significant different overall survival (Wald test $p = 0.086$).

In order to underscore the group of samples with an intermediate molecular profile, and to avoid the over-fitting issues above-mentioned, we chose to keep the results for the 1.5 ratio in the main text, and provide the other results in a supplementary figure (**Fig. S10**).

3. Why did the authors not fit a single RF model, using both methylation and expression as input, instead of having two independent models? Having two models does not leverage the interdependence of the omics layers, requires more parameters and has a lower statistical power

We now provide the rationale for our initial choice of using two independent models (see **page 4** of the **main text** and **pages 25 and 26** of the **Supplementary Methods**), and show in a new **Fig. S9** that using both methylation and expression as inputs leads to similar predictions. More specifically, we now mention in the **Supplementary Methods** that we faced a trade-off between maximizing the number of samples (n) and maximizing the number of features per sample (p). Indeed, for fixed n , as the reviewer mentions, increasing p requires less parameters and leads to higher statistical power. Nevertheless, in our case, because of missing data, increasing p by using both 'omics layers would drastically reduce n , restricting our sample set ($n=158$ and $n=76$ for expression and methylation, respectively) to the set of samples with both layers ($n=51$, including only a single supra-carcinoid). We thus faced the problem of choosing to maximize p at the expense of n or n at the expense of p . Given the existence of very rare entities such as the supra-carcinoids, accurately capturing the diversity of molecular profiles in the training set was our priority, and thus we chose to maximize n . In addition, by maximizing n , we maximized the number of samples with ML predictions, and thus ensured that we would also maximize the power of the subsequent analyses based on the ML results.

In order to check the validity of this initial hypothesis, as suggested by the reviewer, we performed the analysis running the ML algorithm on the restricted set of samples ($n=51$) including both expression and methylation data in the same model and compared the predictions of this model to the combined predictions based on expression and methylation data separately. We found that the predictions (confusion matrix in **Fig. S9**) were similar, with 43/51 samples with both data types predicted similarly in the two models. In addition, we did find that our main finding—the existence of two groups of atypical carcinoid samples, which tended to have a good (red curve) and bad prognosis (pink curve)—still held, but that limited number of samples impeded the statistical analyses. In fact, none of the Cox regression tests were significant, even for the groups with the biggest differences (e.g ML-predicted LCNEC vs ML-predicted typical carcinoid samples); importantly, using the same set of 51 samples, even the survival of the histological types reported by the pathologists were not significantly different (bottom panel). This supports our hypothesis that maximizing p at the expense of n leads to a decrease in power in subsequent analyses due to a smaller sample size. We thus chose to show the results based on the independent models in the main text, and added the suggested analysis combining the two 'omics data together as a supplementary figure (**Fig. S9**).

4. How does a ML model perform when the MOFA factors are used as inputs? How when PCs are used?

We now provide the rationale for our choice of using the expression and methylation features rather than MOFA axes, and show in a new **Fig. S12** that using MOFA factors or principal

components lead qualitatively to similar results than using expression and methylation features directly (see **page 5** of the **main text** and **page 26** of the **Supplementary Methods**).

More specifically, we now mention in the **Supplementary Methods** that because matrix factorization techniques such as MOFA and Principal Component Analyses (PCA) provide low-dimensional approximations of the data, they induce a loss of information. Consequently, if the resulting ML classification is poor, we cannot be sure whether this seemingly bad result is a technical artifact driven by the loss of information due to the dimensionality reduction, or whether this is due to a biologically meaningful mismatch between the histopathological classification and the molecular profiles.

We now show that the suggested analyses based on MOFA factors and the principal components of the PCA lead to similar classification performances as the results presented in the main text **Fig. 1** (confusion matrices in **Fig. S12**). In addition, in the case of MOFA factors, again, atypical carcinoids are stratified into a group with an overall survival similar to that of the LCNEC (in red) and a group with a higher overall survival (in pink), similar to that of the typical carcinoids. When using the principal components, despite a similar trend, the difference in survival between the high- and low-survival groups is not significant. These results show that dimensionality reduction does not lead to increased classification ability, nor does it provide a better explanation of clinical behaviour.

5. Please do not use the term "almost significant", this is non-sensical, statistically speaking

We have made the suggested change throughout the text.

6. When identifying core DE genes: why not simply test group of interest vs rest? Alleviates issues regarding multiple testing in other setting

We now provide the rationale for our choice of method in **page 8** of the **main text** and in **pages 12-13** of the **Supplementary Methods** and include a comparison between our previous statistical analysis and those proposed by the reviewer that shows that our results are conservative (**Fig. S21**). We now mention that, given a focal group A, our goal is to find "core" genes, which expression levels uniquely define A compared to both B and C; a corollary is that we do not want to include genes with similar expression levels between A and B but with large differences between A and C. Our method to achieve this goal is to find DE genes that differentiate A from B (denoted "A vs B"), DE genes that differentiate A from C (denoted "A vs C") and take the intersection of these gene sets [$(A \text{ vs } B) \cap (A \text{ vs } C)$]; this way, we find what gene expressions make the focal group unique. The alternative mentioned by the reviewer, comparing "A vs (B and C)", would select genes that are different between A and the average level of expression of B and C; this would have the unwanted behaviour of including the genes that are strongly DE in the comparison of A vs B, but with similar expression levels in A and C.

In **Fig. S21** we show that our analysis provides conservative results compared to the alternative method proposed. Indeed, core DE genes that we report are almost exclusively a subset of the genes found when comparing the focal group vs the rest. Note that we do not have an increased number of false discoveries due to multiple testing under this setting, because the intersection " $(A \text{ vs } B) \cap (A \text{ vs } C)$ " is a subset of both " $(A \text{ vs } B)$ " and " $(A \text{ vs } C)$ ", so it cannot yield more false discoveries than the comparisons "A vs B" and "A vs C".

7. For the clustering, why not use all MOFA factors. I am not convinced by the citation of the iCluster+ paper as motivation for using 2 factors only: this makes sense in the way the iCluster+ framework is designed in the first place for clustering, but not for MOFA

We now provide more justification on our use of the first latent factors in **page 4** of the **main text** and in **pages 20-21** of the **Supplementary Methods**, and show in a new **Fig. S8** that a clustering using all MOFA factors leads to the exact same clusters. We mention in the main text that MOFA latent factors are ranked from the ones explaining the most variance in gene expression and methylation—expected to present more biological signal—to the ones explaining the least variance—expected to represent more of the noise in the dataset. Thus, we now

importance to the factors most likely to capture the biological variation in the data. 1) We used only the $k-1$ first latent factors, where k is the number of clusters (as in method icluster+ and as in the initial version of the manuscript), and used a consensus clustering approach relying on a k -means algorithm for clustering. Note that we now also further substantiate this choice of number of latent factors for clustering in **page 20** of the **Supplementary Methods** by citing the icluster+ paper from Mo *et al.* (2019), which states that “following a general principle (Hastie, Tibshirani, and Friedman, 2009) for separating g clusters among the n data-points, a rank- k approximation where $k \leq g-1$ is sufficient.” Thus, this choice is more general than the icluster+ method, and applies to all dimensionality reduction techniques that provide such low-rank approximations. 2) We used all five latent factors identified by MOFA, and used a consensus clustering approach relying on a weighted k -means algorithm, where weights correspond to the proportion of variance explained by each latent factor (see details in **page 21** of the **Supplementary Methods**). We show in **Fig. S8** that these two approaches lead to the exact same clusters, both for LLEN and LNET clusters.

8. It's not quite clear from the text how many factors MOFA LLEN identified, only 3? Or were only the first 3 related to clusters? If more, what about the other ones?

We now better describe in **page 4** of the **main text** the MOFA LLEN factors identified. In particular, we now explicitly mention that there was five factors that were above the default 2% variance explained threshold used by MOFA to select the number of factors in the output. In addition, we now mention in **pages 21-22** of the **Supplementary Methods**—as correctly guessed by the reviewer—that among the latent factors, only the first three were associated with either the LLEN clusters ($q = 4.09 \times 10^{-84}$, 8.63×10^{-80} , 0.66, 0.094, 0.24, respectively for latent factors 1 through 5) or the LNET clusters ($q = 5.06 \times 10^{-4}$, 5.99×10^{-47} , 5.12×10^{-46} , 0.15, 0.052, respectively). We now also mention in the discussion that follow up studies are warranted to increase the sample size and potentially reveal additional molecular groups that could be related to other latent factors.

REVIEWERS' COMMENTS:

Reviewer #1 (Remarks to the Author):

The manuscript has been properly revised.

Reviewer #2 (Remarks to the Author):

The authors have addressed my concerns in full.